# Depleting chemoresponsive mitochondrial fission mediator DRP1 does not mitigate sarcoma resistance

Karolina Borankova[1,2], Matyas Solny[1], Maria Krchniakova[1], Jan Skoda[1,2]

**Specific patterns of mitochondrial dynamics have been repeatedly reported to promote drug resistance in cancer. However, whether targeting mitochondrial fission– and fusion–related proteins could be leveraged to combat multidrug-resistant pediatric sarcomas is poorly understood. Here, we demonstrated that the expression and activation of the mitochondrial fission mediator DRP1 are affected by chemotherapy exposure in common pediatric sarcomas, namely, rhabdomyosarcoma and osteosarcoma. Unexpectedly, decreasing DRP1 activity through stable DRP1 knockdown neither attenuated sarcoma drug resistance nor affected growth rate or mitochondrial network morphology. The minimal impact on sarcoma cell physiology, along with the up-regulation of fission adaptor proteins (MFF and FIS1) detected in rhabdomyosarcoma cells, suggests an alternative DRP1-independent mitochondrial fission mechanism that may efficiently compensate for the lack of DRP1 activity. By exploring the upstream mitophagy and mitochondrial fission regulator, AMPKα1, we found that markedly reduced AMPKα1 levels are sufficient to maintain AMPK signaling capacity without affecting chemosensitivity. Collectively, our findings challenge the direct involvement of DRP1 in pediatric sarcoma drug resistance and highlight the complexity of yet-to-be-characterized noncanonical regulators of mitochondrial dynamics.**

## Introduction

Pediatric sarcomas are a group of distinct mesenchymal solid tumors that arise either in bone or in mesenchymal soft tissue and account for ~20% of childhood nonhematologic malignancies (1). Therapy failure because of the development of multidrug resistance, resulting in a dismal prognosis, is frequent in the most prevalent pediatric sarcomas, such as rhabdomyosarcoma and osteosarcoma; the 5-yr survival rates of their high-grade forms reach less than 50% (2, 3, 4). Although specific patterns of mitochondrial dynamics have emerged as important contributors to multidrug resistance in various cancers (5, 6, 7), little is known about the role of mitochondrial dynamics in pediatric sarcomas.

Mitochondrial dynamics is an umbrella term for mitochondrial adaptations consisting of mitochondrial fission and fusion, which shape the mitochondrial network, and mitochondrial biogenesis and mitophagy, a selective mitochondrial degradation process that fine-tunes mitochondrial quantity and quality (7). Deregulated mitochondrial dynamics have been implicated in tumorigenesis as the expression of mitochondrial dynamics–related proteins differs between tumor and adjacent non-tumor tissues (8, 9, 10, 11, 12) and can be applied to predict patients' outcomes (11, 13, 14, 15, 16, 17). Furthermore, enhanced mitochondrial fission (18, 19, 20, 21) and mitophagy (20, 21, 22, 23) have been shown to promote a cancer stem-like phenotype in various tumor types. Consistently, mitochondrial fission (17, 24, 25, 26, 27) and fusion (28, 29, 30, 31, 32) and mitophagy (33, 34, 35) have been reported to mediate multidrug resistance, which can be mitigated by both pharmacological inhibition and modulation of the expression of mitochondrial dynamics–related proteins. Together, these reports suggest mitochondrial dynamics as a promising therapeutic target for overcoming resistance to standard treatment protocols. However, discrepancies between individual reports and contradictory results in various models indicate vast complexity and high context dependency, hindering translation into the clinic. For instance, shifting mitochondrial dynamics toward fission (24, 28, 29, 30, 31, 32) or fusion (9, 25, 26, 27, 36) has been reported to suppress resistance to cancer therapy. Therefore, further research is needed to elucidate the tumor type–specific role of mitochondrial dynamics in multidrug resistance and its implications for cancer.

To investigate the potential of targeting mitochondrial dynamics for sarcoma therapy, we analyzed the modulation of mitochondrial fission– and fusion–related proteins after drug exposure in the most common pediatric sarcomas, rhabdomyosarcoma and osteosarcoma, which have been largely overlooked in mitochondrial research thus far. Notably, we showed that the expression and activating phosphorylation of the mitochondrial fission mediator dynamin-related protein 1 (DRP1) are modulated in sarcoma cells upon chemotherapy exposure. However, depletion of DRP1 or the mitochondrial dynamics upstream regulator AMP-activated protein

---

[1]Department of Experimental Biology, Faculty of Science, Masaryk University, Brno, Czech Republic  [2]International Clinical Research Center, St. Anne's University Hospital, Brno, Czech Republic

Correspondence: jan.skoda@sci.muni.cz

kinase catalytic subunit alpha 1 (AMPKα1) strikingly did not affect sarcoma cell physiology or attenuate multidrug resistance. These results provide new mechanistic evidence of noncanonical DRP1-independent mitochondrial fission, revealing the potential of sarcoma cells to escape DRP1-targeted anticancer therapies.

# Results

### The mitochondrial fission mediator DRP1 predicts outcomes in transcriptomic cohorts and exhibits enhanced activation in post-therapy pediatric sarcoma-derived cells

To explore the therapeutic potential of mitochondrial dynamics–related proteins (Fig 1A), we first analyzed the expression of their respective genes in relation to patient survival using publicly available rhabdomyosarcoma and osteosarcoma transcriptomics datasets. This analysis revealed that the expression of the gene encoding the mitochondrial fission mediator DRP1 *DNM1L* could stratify both the rhabdomyosarcoma and osteosarcoma cohorts into groups with significant differences in survival (Fig 1B). Interestingly, high *DNM1L* expression was associated with a poor prognosis in patients with osteosarcoma but predicted better outcomes in patients with rhabdomyosarcoma, suggesting a possible sarcoma subtype–dependent role for mitochondrial fission in the therapeutic response.

To elucidate whether initial drug exposure affects mitochondrial dynamics, we included rhabdomyosarcoma and osteosarcoma cell lines derived from both therapy-naive (pre-therapy) and relapsed/refractory (post-therapy) tumors. Immunoblotting revealed differential expression of mitochondrial fission– and fusion–related proteins across a panel of tested sarcoma cells (Fig 1C and D). Notably, the activating phosphorylation of the mitochondrial fission mediator DRP1 at serine 616 (40) was strongly up-regulated in two out of three post-therapy cell lines, indicating that enhanced DRP1 activation might be associated with a potential mechanism of adaptation to therapy-induced stress. However, to draw more robust conclusions, it will be necessary to expand the panel of tested cell lines in follow-up studies.

Interestingly, post-therapy osteosarcoma Saos-2 cells showed up-regulated levels of both mitochondrial fission– and fusion–related proteins despite maintaining relatively low mitochondrial mass, as assessed by the expression of the translocase of outer mitochondrial membrane 20 (TOMM20), which serves as a proxy for mitochondrial content. In contrast, the post-therapy RD rhabdomyosarcoma cell line exhibited increased levels solely in the mitochondrial fission mediator DRP1, along with strong up-regulation of its activating phosphorylation. Given these potential therapy-related patterns, we aimed to analyze whether these differences are reflected in the modulation of mitochondrial fission– and fusion–related proteins in response to chemotherapy exposure.

### Autophagy inhibition suppresses the drug-induced activating phosphorylation of DRP1 but does not mitigate rhabdomyosarcoma drug resistance

Mitochondrial fission mediated by DRP1 not only changes mitochondrial morphology but also separates damaged mitochondria into autophagosomes for autophagic degradation (41). The ongoing autophagic flux can be pharmacologically blocked by bafilomycin A1 (Fig 2A). Compared with other sarcoma-derived cell lines, rhabdomyosarcoma RD cells, which exhibit high levels of DRP1 expression and its activating phosphorylation, exhibited greater vulnerability to bafilomycin A1–mediated inhibition of autophagy (Fig 2B). Considering the high sensitivity to autophagy inhibition and the increased levels of DRP1-activating phosphorylation, we assumed that high autophagic flux, which indicates high mitophagy activity, is crucial for the survival of RD cells. Hence, mitochondrial fission– and fusion–related protein levels were analyzed in RD cells after exposure to drugs at half-maximal inhibitory concentrations (IC$_{50}$ values, dose–response curves detailed in Fig S1) and autophagy inhibition by a sublethal dose of bafilomycin A1. Standard chemotherapy drugs commonly used for pediatric sarcoma—cisplatin (42, 43), doxorubicin (43, 44), and topotecan (45, 46), all of which have predominantly genotoxic effects, and a microtubule polymerization inhibitor, vincristine (44, 47)—were used in these experiments. Exposure to chemotherapeutics in vitro increased the level of DRP1-activating phosphorylation, and this effect was diminished by concurrent autophagy inhibition with bafilomycin A1 (Fig 2C and D). This finding suggested that the increase in DRP1-activating phosphorylation at S616 might serve as an adaptation to chemotherapy-induced stress and that inhibiting autophagy, thereby mitophagy, prevents this response.

Unexpectedly, other mitochondrial fission– and fusion–related proteins in RD cells were generally unaffected by drug exposure (Figs 2C and S2). However, the stress-responsive AMP-activated protein kinase (AMPK), known to promote both autophagy and mitochondrial fission (48), was activated by cisplatin and doxorubicin treatment (Fig 2C and D). Considering that both cisplatin and doxorubicin enhanced autophagic flux in RD cells, as indicated by the increase in the isoform II of microtubule-associated protein 1 light chain 3A and 3B (LC3AB-II) upon bafilomycin A1–mediated inhibition of autophagy (Fig 2C and D), we speculated that, in line with reports in other cell types (33, 49, 50), autophagy might protect rhabdomyosarcoma cells against these genotoxic drugs. To test this hypothesis, we treated rhabdomyosarcoma cells with cisplatin and doxorubicin as single agents or in combination with the autophagy inhibitor bafilomycin A1 or the autophagy inducer rapamycin. Rapamycin is known to promote autophagy by inhibiting mammalian target of rapamycin complex 1, a central controller of cell metabolism and a key negative regulator of the ULK1 autophagy initiation complex (51). Indeed, rapamycin reduced sensitivity to cisplatin and doxorubicin in both bafilomycin A1–sensitive (RD) and bafilomycin A1–resistant (NSTS-46 and NSTS-11) rhabdomyosarcoma cells (Figs 3 and S3), suggesting that enhanced autophagy might serve as a mechanism of multidrug resistance in rhabdomyosarcoma. However, combined treatment with bafilomycin A1 did not enhance the effects of these chemotherapy drugs, even in autophagy inhibition–sensitive RD cells (Fig S4). This finding indicates that the autophagy-independent effects of mammalian target of rapamycin complex 1 inhibition promote the resistance of rhabdomyosarcoma cells to cisplatin and doxorubicin. These results additionally contradict the cytoprotective role of drug-induced activation of DRP1, which was diminished by bafilomycin A1.

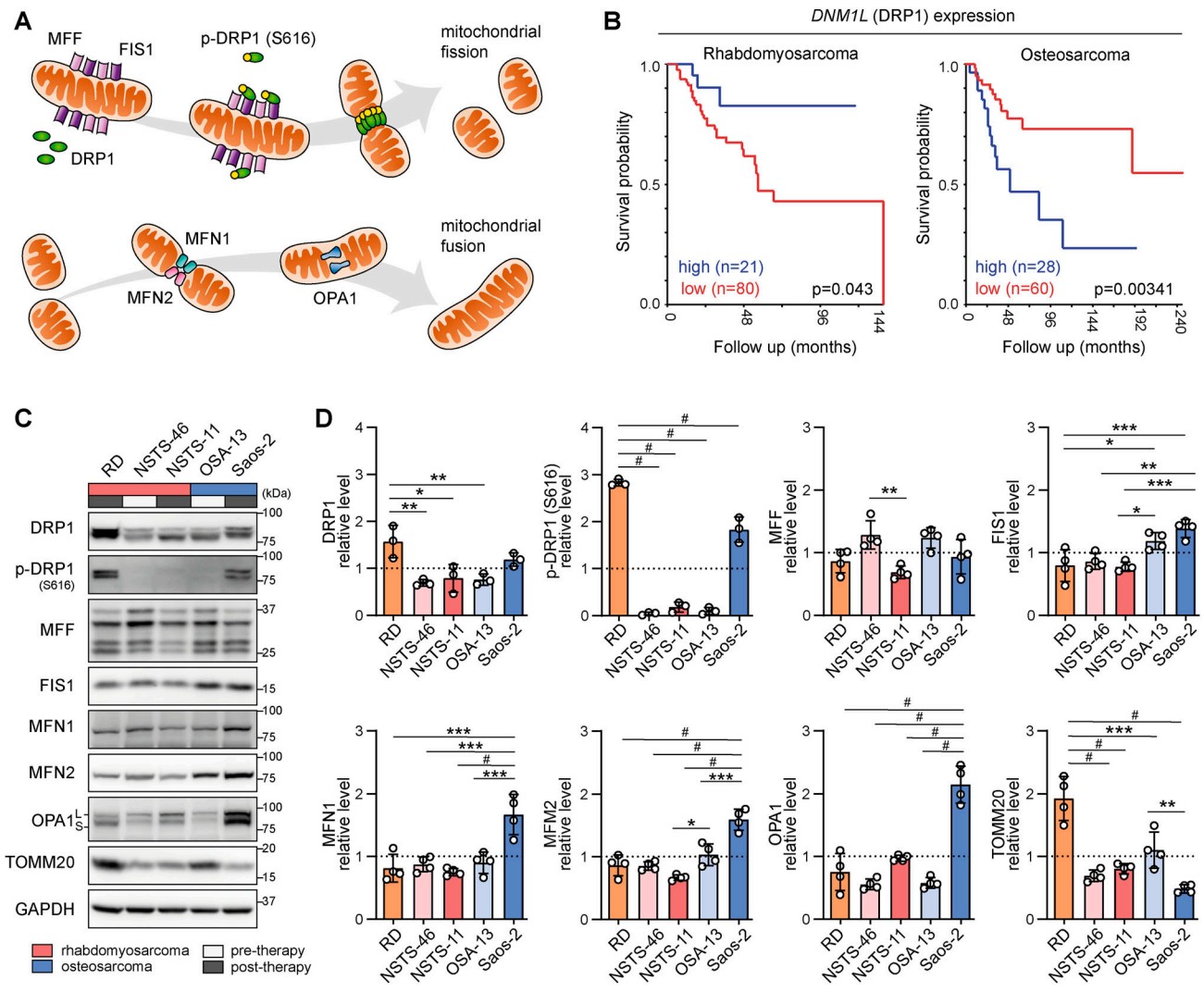

**Figure 1. The mitochondrial fission mediator DRP1 predicts outcomes in transcriptomic cohorts and shows enhanced activation in post-therapy pediatric sarcoma-derived cells.**

**(A)** An overview of mitochondrial fission– and fusion–related proteins analyzed in the study. The mitochondrial fission mediator DRP1, which is activated by the phosphorylation at S616, forms a contractile ring on the mitochondrial surface by interacting with adaptor proteins such as mitochondrial fission factor and mitochondrial fission 1 protein (FIS1), leading to mitochondrial constriction and fission. Mitofusin-1 (MFN1) and mitofusin-2 (MFN2) mediate fusion of the outer mitochondrial membrane, whereas OPA1 mitochondrial dynamin–like GTPase (OPA1) mediates fusion of the inner mitochondrial membrane. For further details, including additional components of the mitochondrial fission and fusion machinery, please refer to Reference 37. **(B)** Kaplan–Meier analysis of the overall survival of patients stratified by the expression of the DRP1 gene *DNML1* was performed via the R2: Genomics Analysis and Visualization Platform. The analysis of rhabdomyosarcoma (38) and osteosarcoma (39) transcriptomic datasets was performed using scan cutoff mode with a minimal group size of 20 patients. **(C, D)** Western blotting (C) and densitometric analysis (D) of mitochondrial fission– and fusion–related proteins across a panel of rhabdomyosarcoma and osteosarcoma cell lines derived from both therapy-naive (pre-therapy) and relapsed/refractory (post-therapy) tumors. **(D)** Statistical significance was determined by one-way ANOVA followed by Tukey's multiple comparisons test, *$P < 0.05$, **$P < 0.01$, ***$P < 0.001$, #$P < 0.0001$.

Source data are available for this figure.

## Modulation of the mitochondrial fission machinery in sarcoma cells upon chemotherapy exposure reflects the initial DRP1 status

Analysis of mitochondrial fission– and fusion–related proteins in bafilomycin-resistant rhabdomyosarcoma cells (Fig 4A and C and S5) and osteosarcoma cells (Figs 4B and C and S5) exposed to $IC_{50}$ doses of chemotherapy drugs (as detailed in Fig S1) revealed that the drug-induced modulation of DRP1 depended on the initial status of DRP1. In Saos-2 osteosarcoma cells with increased levels of DRP1-activating

phosphorylation (Fig 1C and D), exposure to genotoxic drugs increased the levels of this p-DRP1(S616) form (Fig 4B and C). In contrast, in sarcoma cells without detectable p-DRP1(S616), chemotherapy exposure further suppressed the levels of mitochondrial fission machinery proteins (Fig 4A and C). A decrease in total DRP1 was detected in NSTS-11 cells, whereas NSTS-46 and OSA-13 cells—expressing the highest basal levels of the DRP1 adaptor, mitochondrial fission factor (MFF) (Fig 1C and D)—showed prevalent down-regulation of MFF in response to the tested drugs (Fig 4A–C). Interestingly, the levels of

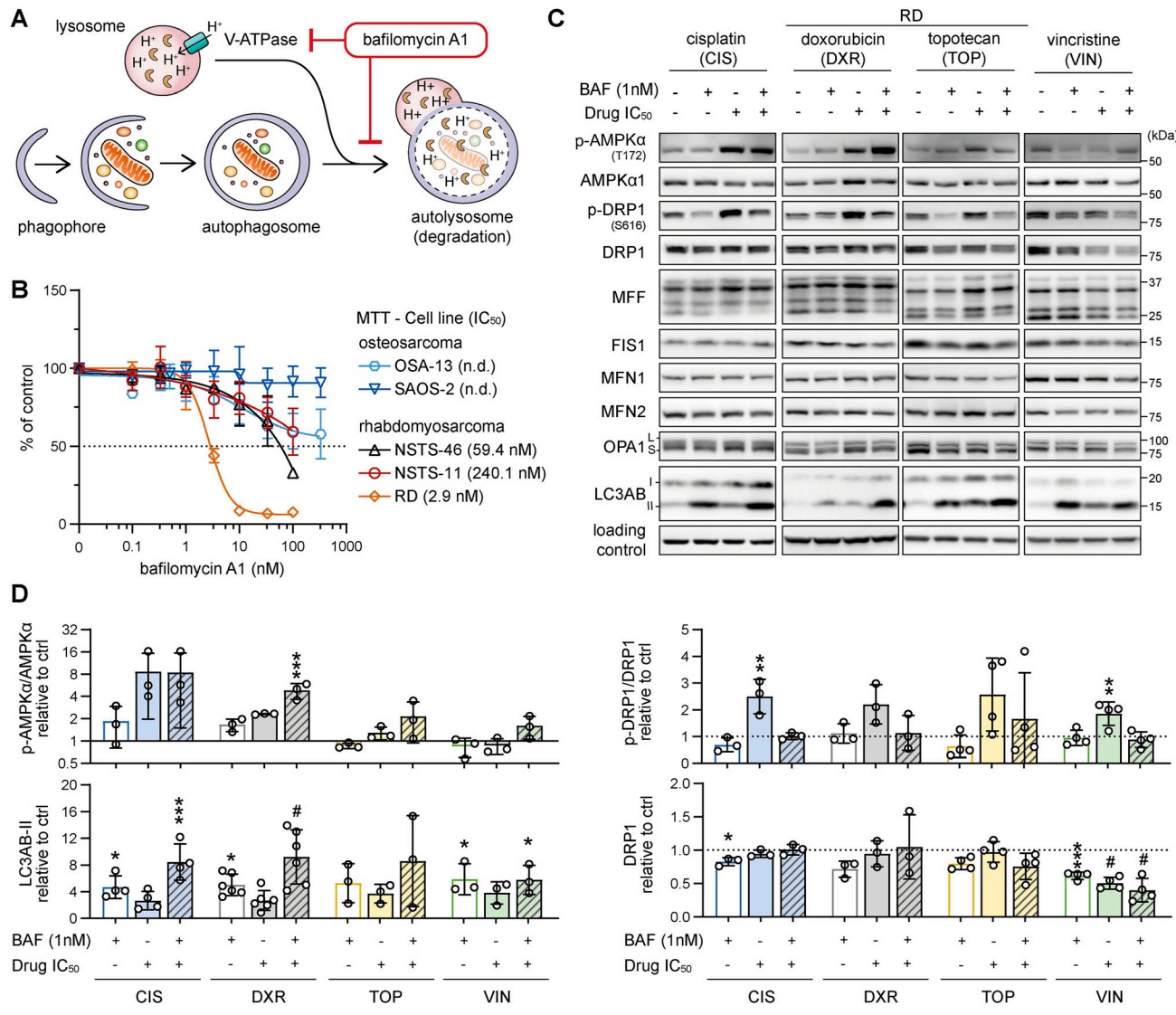

**Figure 2. Bafilomycin A1–mediated autophagy inhibition suppresses the drug-induced activating phosphorylation of DRP1.**
**(A)** An overview of the mechanism of bafilomycin A1–mediated autophagy inhibition. Cellular components marked for autophagic degradation are engulfed by the phagophore, leading to autophagosome formation. The autophagosome content is degraded because of fusion with acidified lysosomes. Bafilomycin A1 inhibits both lysosomal acidification and lysosome-autophagosome fusion, thereby inhibiting autophagic flux. **(B)** MTT viability assay analysis after 72 h of treatment revealed variable sensitivity to bafilomycin A1 in a panel of rhabdomyosarcoma and osteosarcoma cell lines derived from both therapy-naive (pre-therapy) and relapsed/refractory (post-therapy) tumors. The respective $IC_{50}$ values are indicated in brackets; n.d. indicates that the $IC_{50}$ was not measurable because of marked resistance. The data points represent the mean ± SD, biological: n = 3, technical: n = 3. **(C, D)** Western blotting (C) and densitometric analysis (D) of proteins related to mitochondrial fission/fusion and autophagy in RD rhabdomyosarcoma cells exposed to 1 nM bafilomycin A1 (BAF), chemotherapy drugs at the $IC_{50}$ doses, or combinations of BAF with chemotherapy drugs for 72 h. Normalized protein levels are plotted relative to the untreated controls. Mean ± SD. BAF, bafilomycin A1; CIS, cisplatin; DXR, doxorubicin; TOP, topotecan; VIN, vincristine. Densitometric analysis of MFF, FIS1, MFN1, MFN2, and OPA1 is provided in Fig S2. **(D)** Statistical significance was determined by one-way ANOVA followed by Tukey's multiple comparisons test, *P < 0.05, **P < 0.01, ***P < 0.001, #P < 0.0001.
Source data are available for this figure.

mitochondrial fission 1 protein (FIS1), another DRP1 adaptor protein, did not show any clear trend upon drug exposure (Figs 4A and B and S5), suggesting that only a subset of mitochondrial fission–related proteins are affected by drug treatment. Conversely, the expression of fusion-related proteins was mostly unaffected after treatment (Figs 4A–C and S5), with one exception: mitofusin-1 (MFN1) was down-regulated in response to topotecan in NSTS-11 rhabdomyosarcoma cells and both osteosarcoma cell lines. Although these results exclude

the role of MFN1 in sarcoma multidrug resistance, further research might elucidate its potential involvement in regulating selective resistance to topotecan. In addition to mitofusin-2 (MFN2), the mediator of inner mitochondrial membrane fusion, mitochondrial dynamin-like GTPase OPA1 (OPA1), remained intact in drug-exposed sarcoma cells (Figs 4A and B and S5), demonstrating that chemotherapy did not activate the mitochondrial stress sensors OMA1 zinc metallopeptidase (OMA1) and ATP-dependent zinc metalloprotease YME1L1 (YME1L1),

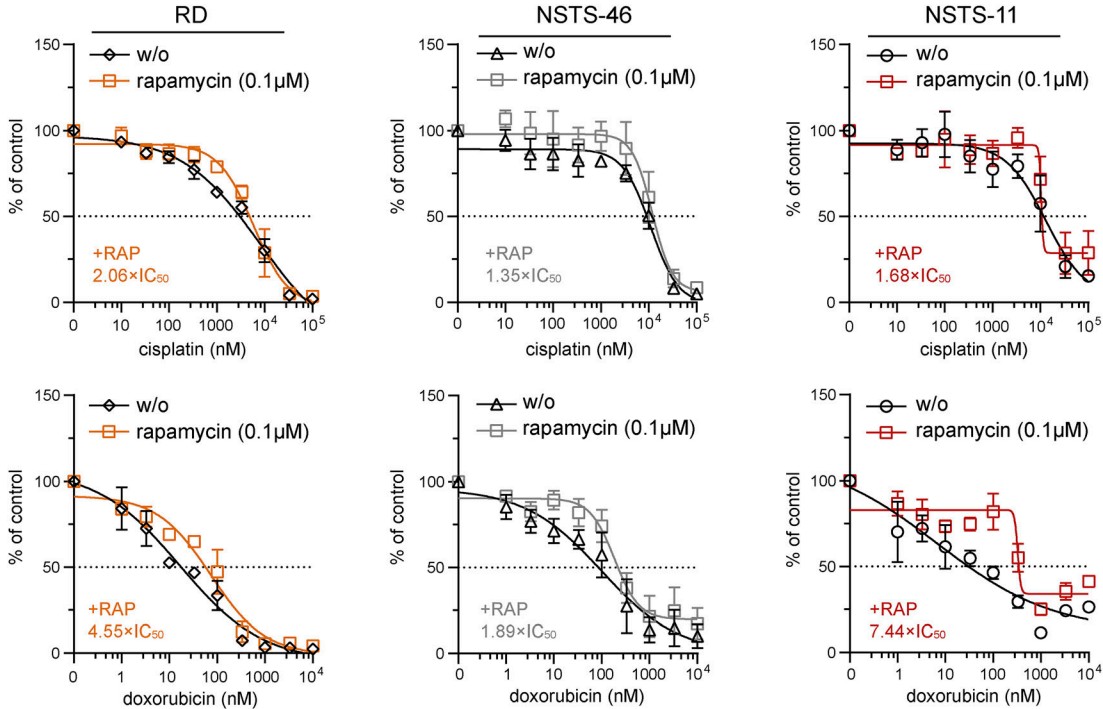

**Figure 3. The mTOR inhibitor rapamycin reduced sensitivity to cisplatin and doxorubicin in both bafilomycin A1–sensitive and bafilomycin A1–resistant rhabdomyosarcoma cells.**
MTT viability assay analysis of bafilomycin A1–sensitive (RD) and bafilomycin A1–resistant (NSTS-46, NSTS-11) rhabdomyosarcoma cells exposed to the indicated concentrations of cisplatin and doxorubicin alone or in the presence of 0.1 $\mu$M rapamycin for 72 h. The rapamycin-induced increase in the $IC_{50}$ is indicated as the ratio of the $IC_{50}$ determined for the combination of rapamycin and chemotherapy drugs to the $IC_{50}$ determined for chemotherapy drugs alone. The data points represent the mean ± SD, biological: n ≥ 3, technical: n = 3. Extended analysis of determined $IC_{50}$ values is provided in Fig S3.

which cleave OPA1 into its short forms (S-OPA1) under mitochondrial stress. Unexpectedly, we observed preferential down-regulation of these mitochondrial stress sensors in drug-exposed sarcoma cells (Fig S6), suggesting that OMA1 and YME1L1 may have OPA1 processing–independent functions in sarcoma chemotherapy adaptation; similarly, OMA1 was recently reported to induce a metabolic shift upon DNA damage (52).

### Stable knockdown of DRP1 does not impair sarcoma cell physiology or decrease drug resistance

Collectively, the analysis of mitochondrial dynamics after drug exposure suggested that mitochondrial fission is a context-dependent mechanism that fine-tunes sarcoma drug resistance. Since sarcoma survival was stratified by DRP1 expression (Fig 1B) and up-regulated DRP1-activating phosphorylation at S616 was associated with the post-therapy state (Fig 1C and D), with exposure to chemotherapeutics further increasing this phosphorylation (Figs 2C and D and 4A–C), we hypothesized that DRP1 activity is involved in the mechanism that protects sarcoma cells against chemotherapy-induced stress. To test this hypothesis, we established cells with stable DRP1 knockdown by lentiviral transduction of constructs encoding target-specific shRNAs under the control of a constitutive promoter. Characterization of single-cell–derived clones from DRP1 knockdown (shDRP1) and control (shCTRL) cells confirmed the depletion of DRP1 and demonstrated a marked decrease in its activating phosphorylation form in both

rhabdomyosarcoma and osteosarcoma models (Figs 5A and E and S7). Despite significantly decreased DRP1-activating phosphorylation, the cell cycle progression and growth rates of rhabdomyosarcoma (Figs 5B and S8) and osteosarcoma shDRP1 clones (Figs 5F and S9) were very similar to those of their control counterparts, suggesting that DRP1 is not required for sarcoma cell proliferation. Strikingly, DRP1 down-regulation did not mitigate chemoresistance in either rhabdomyosarcoma or osteosarcoma models. A comparison of $IC_{50}$ values determined in shCTRL and shDRP1 clones did not reveal any significant differences in resistance to standard chemotherapy (Figs 5C and G, S10A, and S11A) or targeted inhibitors (Figs 5D and H, S10B, and S11B), except for slightly reduced sensitivity to cisplatin and vincristine detected in rhabdomyosarcoma but not osteosarcoma shDRP1 clones (Figs 5C and G, S10, and S11). Hence, we hypothesized that DRP1 plays a dispensable role in sarcoma cell physiology, which is further supported by the similar sensitivities of the shCTRL and shDRP1 clones to autophagy inhibitors (bafilomycin A1 and chloroquine) and the respiratory complex I inhibitor phenformin (Fig S12).

To further elucidate the very limited impact of DRP1 knockdown on sarcoma cell physiology, we aimed to assess whether DRP1 depletion shifted the balance between mitochondrial fission and fusion. Microscopically, the morphology of the mitochondrial network was unchanged in the shDRP1 clones compared with the control clones in both rhabdomyosarcoma (Figs 6A, S13, and S14) and osteosarcoma (Figs 7A, S15, and S16). Considering the near absence of the activated p-DRP1(S616) form in shDRP1 clones, these

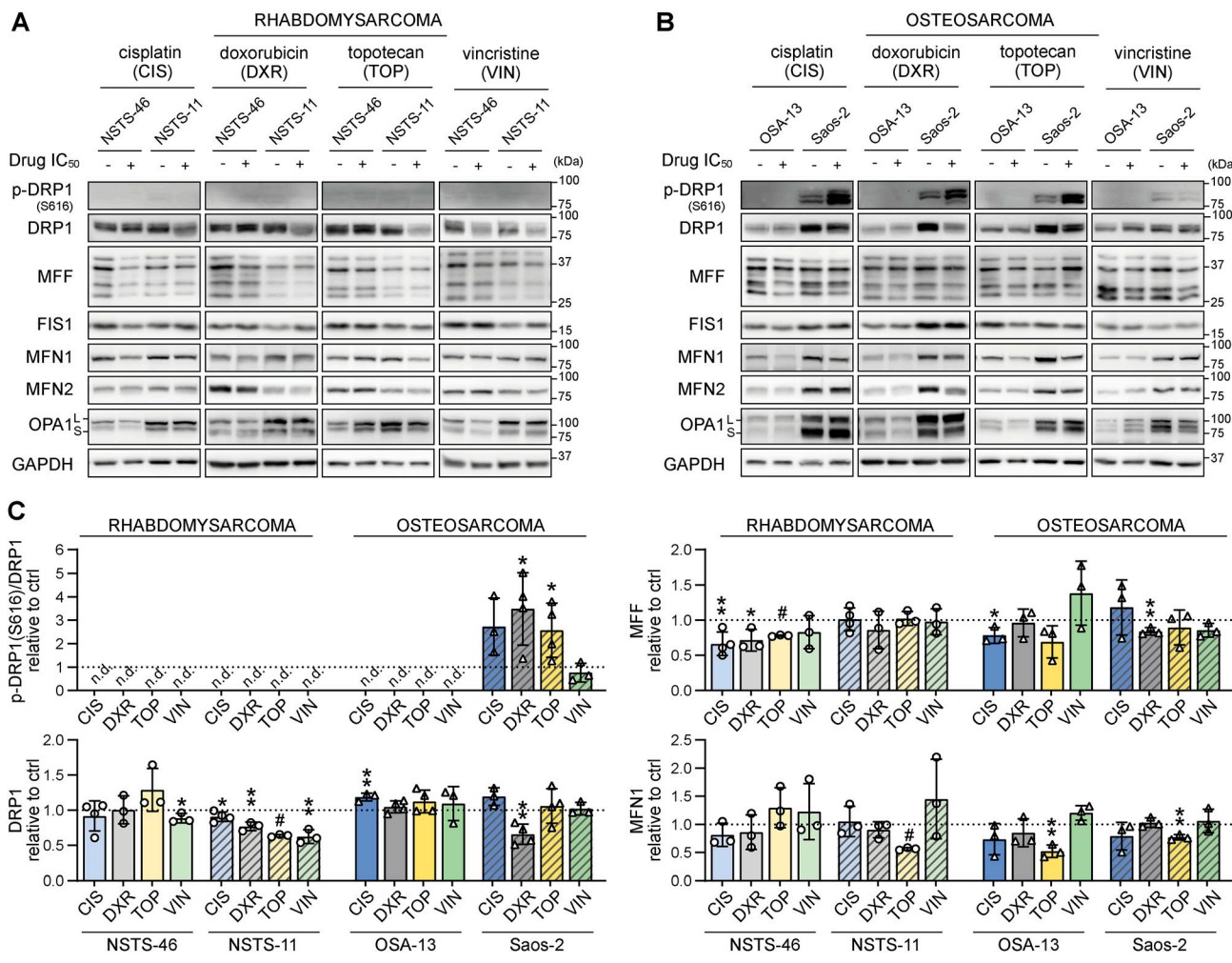

**Figure 4. Modulation of the mitochondrial fission machinery upon chemotherapy exposure reflects the initial DRP1 status in sarcoma cells.**
**(A, B, C)** Western blotting (A, B) and densitometric analysis (C) of mitochondrial fission– and fusion–related proteins in rhabdomyosarcoma (A, C) and osteosarcoma (B, C) cells exposed to IC$_{50}$ doses of chemotherapy for 72 h. Normalized protein levels are plotted relative to the untreated controls as the mean ± SD. CIS, cisplatin; DXR, doxorubicin; TOP, topotecan; VIN, vincristine. Densitometric analysis of FIS1, MFN2, and OPA1 is provided in Fig S5. **(C)** Statistical significance was determined by an unpaired two-tailed $t$ test, $*P < 0.05$, $**P < 0.01$, $\#P < 0.0001$; n.d., not detected.
Source data are available for this figure.

results indicate that DRP1-independent mitochondrial fission compensates for the lack of mitochondrial membrane scission mediated by DRP1. In the rhabdomyosarcoma model, immunoblotting revealed up-regulation of the mitochondrial outer membrane adaptor proteins MFF and FIS1 (Figs 6B and S17), which might contribute to alternative DRP1-independent mitochondrial fission (53). Nevertheless, further research is needed to elucidate the mechanisms compensating for DRP1 inactivation because no modulation of mitochondrial fission– or fusion–related proteins was detected in osteosarcoma shDRP1 clones (Figs 7B and S18).

## Rhabdomyosarcoma drug resistance and stress-induced modulation of mitochondrial dynamics proteins are independent of the upstream mitochondrial fission and autophagy inducer AMPKα1

Given the identified capacity of sarcoma cells to compensate for DRP1, which is presumably a key component of the mitocho

ndrial fission machinery, we hypothesized that targeting upstream regulators might be necessary to overcome drug resistance. Therefore, we established rhabdomyosarcoma clones with stable knockdown of AMPKα1 (Figs 8A and S19), an isoform of the catalytic subunit of stress-responsive AMPK that promotes both autophagy and mitochondrial fission upon activation (48). This AMPKα1 isoform, which is frequently amplified in tumors (54, 55), has been reported to enhance resistance against genotoxic agents (56, 57). Notably, we demonstrated that cisplatin and doxorubicin treatments activated AMPK in RD cells (Fig 2C and D). As expected, no difference in the growth rate was detected between the control (shCTRL) and AMPKα1 knockdown (shAMPKα1) clones (Fig 8B) as AMPK activity is induced solely under stress conditions. However, the down-regulation of AMPKα1 did not decrease resistance to standard chemotherapy (Figs 8C and S20) or targeted inhibitors (Figs 8D and S20). Consistent with these findings, sensitivity to autophagy inhibitors (Fig S21) and the respiratory complex I

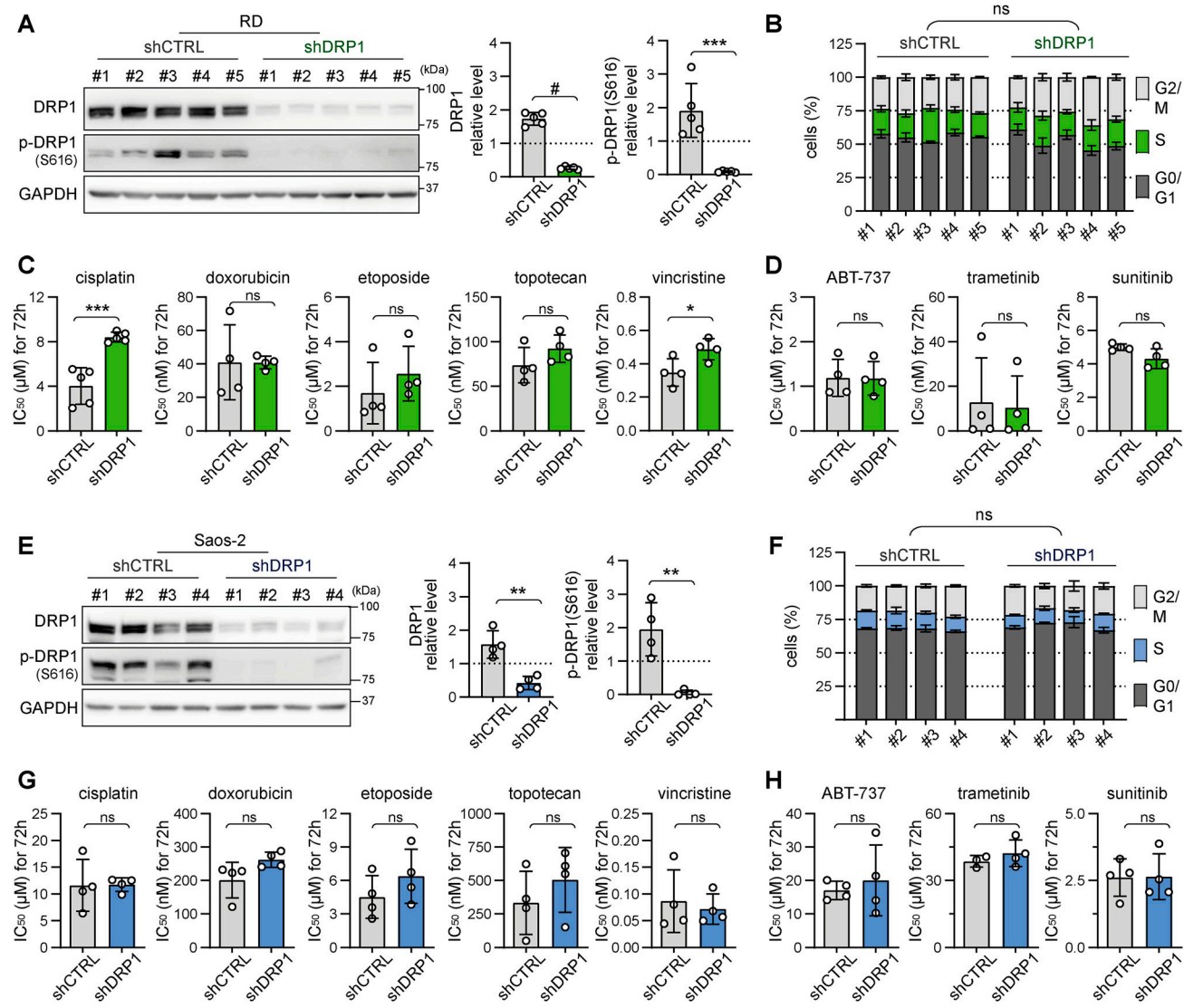

**Figure 5. Stable knockdown of DRP1 does not impair sarcoma cell growth or enhance drug sensitivity.**
**(A, E)** Western blotting and densitometric analysis confirmed the efficiency of DRP1 knockdown in rhabdomyosarcoma RD (A) and osteosarcoma Saos-2 (E) cell clones. The mean normalized protein levels detected in the control (shCTRL) and DRP1–down-regulated (shDRP1) cell clone groups are plotted as the mean ± SD. Densitometric analysis of biological replicates of individual cell clones is provided in Fig S7. **(B, F)** Flow cytometry analysis of the cell cycle of individual RD (B) and Saos-2 (F) shCTRL and shDRP1 cell clones. The data are plotted as the mean ± SD of the cell proportions in the G0/G1, S, and M phases of the cell cycle; ns, not significant, refers to the statistical analyses provided in Figs S8 and S9. **(C, D, G, H)** The chemoresistance of control and DRP1–down-regulated sarcoma cells was compared using the drug IC$_{50}$. The data are plotted as the mean ± SD of the IC$_{50}$ values determined for individual RD (C, D) and Saos-2 (G, H) shCTRL and shDRP1 cell clones. Dose–response curves for standard chemotherapy drugs (C, G) and targeted inhibitors (D, H) are provided in Figs S10 and S11. Statistical significance was determined by an unpaired two-tailed t test, *$P <$ 0.05, **$P <$ 0.01, ***$P <$ 0.001, #$P <$ 0.0001; ns, not significant.
Source data are available for this figure.

inhibitor phenformin (Fig S21) was not affected by AMPKα1 knockdown.

Since drug resistance was unaffected by AMPKα1 knockdown, we aimed to investigate whether the down-regulation of AMPKα1 modified mitochondrial dynamics downstream of activated AMPK. Unexpectedly, after 72 h of exposure to a well-established AMPK activator, phenformin, no significant differences in the levels of mitochondrial fission– or fusion–related proteins were detected between the shCTRL and shAMPKα1 clones (Fig 8E–G and S22). In addition, upon nutrient deprivation combined with phenformin

treatment, AMPKα1 knockdown cells exhibited a rapid increase in the level of activated AMPK, p-AMPKα(T172), which was comparable to that of their control counterparts (Fig S23). Another AMPKα isoform, AMPKα2, might compensate for AMPKα1 depletion, potentially explaining the observed AMPK activation. However, this effect was excluded by immunoblotting, which demonstrated that AMPKα2 expression was undetectable in both the shCTRL and shAMPKα1 clones (Fig S19), even upon phenformin-mediated AMPK activation (Figs S22 and S23). Collectively, our data reveal that very low levels of AMPK are sufficient to maintain AMPK signaling and

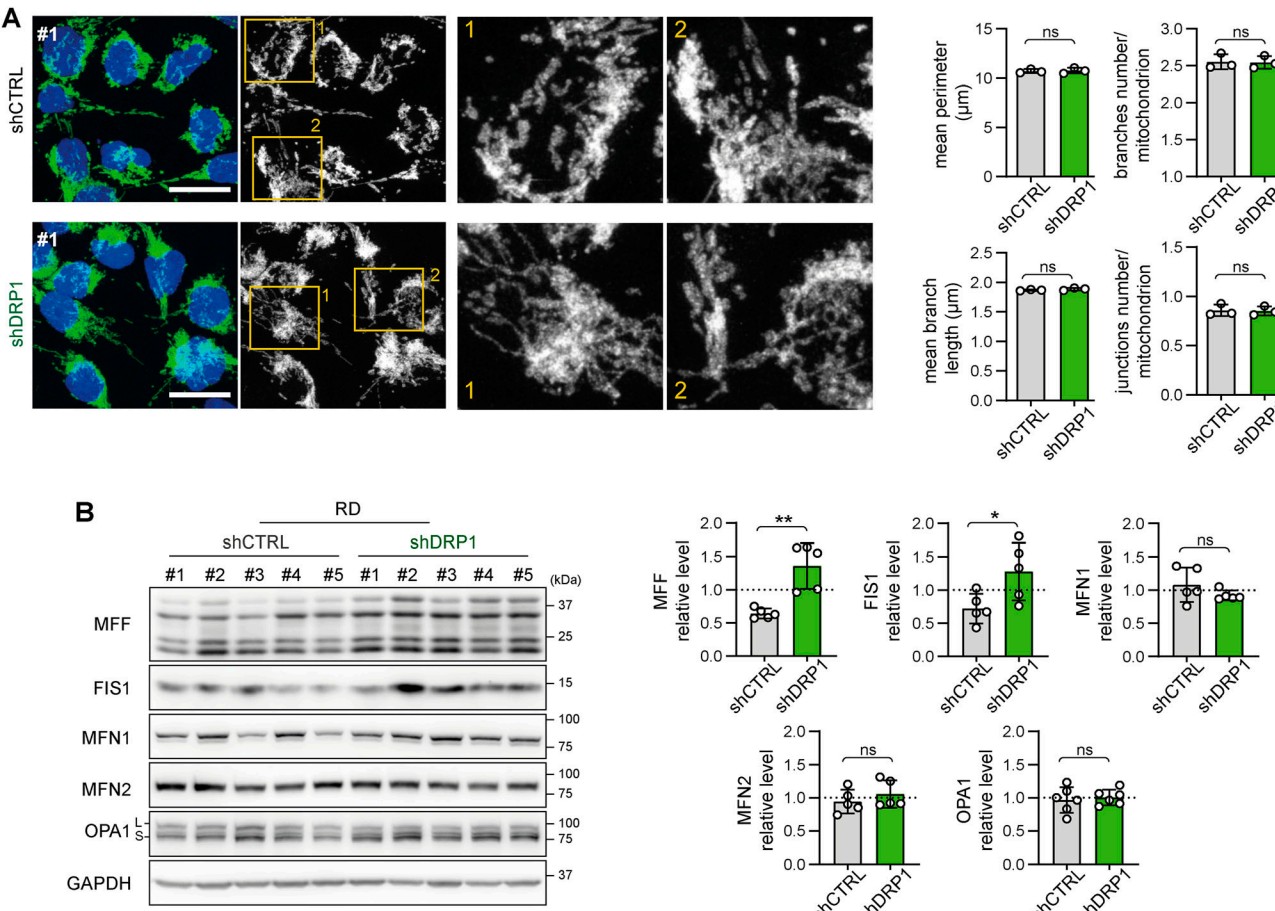

**Figure 6.  Depletion of DRP1 does not affect the mitochondrial network morphology in RD rhabdomyosarcoma cells.**
**(A)** Mitochondrial morphology, as visualized by immunofluorescence staining of TOMM20 (green), was not altered by DRP1 down-regulation. Representative merged images (green: TOMM20, blue: nuclei) and grayscale images (mitochondria) of maximum intensity projections of confocal microscopy Z-stacks are shown for individual RD control (shCTRL) and DRP1–down-regulated (shDRP1) cell clones. White bar: 20 $\mu$m. Images of mitochondria from other analyzed clones are provided in Fig S13. Image analysis using the ImageJ plug-in tool Mitochondria Analyzer showed that parameters describing the mitochondrial network morphology did not significantly differ between the shCTRL and shDRP1 clones. The mean parameters determined for the shCTRL and shDRP1 clone groups are presented as the mean ± SD. Analysis of biological replicates of individual cell clones is provided in Fig S14. **(B)** Western blotting and densitometric analysis of mitochondrial fission– and fusion–related proteins in RD shCTRL and shDRP1 cell clones. The mean normalized protein levels detected in the shCTRL and shDRP1 clone groups are plotted relative to the average levels as the mean ± SD. Densitometric analysis of biological replicates of individual cell clones is provided in Fig S17. Statistical significance was determined by an unpaired two-tailed *t* test, *$P$ < 0.05, **$P$ < 0.01; ns, not significant.
Source data are available for this figure.

mitochondrial responses after AMPKα1 knockdown. Thus, models with concurrent knockout of both AMPKα isoforms should provide a definite answer as to whether AMPK signaling might promote rhabdomyosarcoma multidrug resistance by regulating mitochondrial dynamics.

## Discussion

Drug resistance is a significant factor contributing to the failure of cancer therapy. However, effective treatment protocols that would prevent the induction of acquired resistance remain elusive. Recently, enhanced mitochondrial dynamics have emerged as a common vulnerability of drug-resistant and aggressive cancer cells, and the mitochondrial fission mediator DRP1 has been repeatedly

suggested as a potential therapeutic target (9, 11, 17, 18, 27). High expression of DRP1 and its adaptors has been associated with poor survival in several adult tumor types, including breast carcinoma (13, 58), nasopharyngeal carcinoma (17), lung carcinoma (11), and hepatocellular carcinoma (14, 15, 59, 60). Consistent with these findings, we found that DRP1 expression can be used to stratify pediatric sarcoma patients according to survival, albeit in a sarcoma subtype–dependent manner (Fig 1B). High DRP1 expression is associated with poor survival in patients with osteosarcoma but predicts better outcomes in patients with rhabdomyosarcoma, suggesting that mitochondrial fission plays a tumor type–dependent role in the therapeutic response. This finding in rhabdomyosarcoma aligns with studies that revealed an association between low expression of DRP1 adaptors and a poor prognosis in tongue squamous cell carcinoma patients (24, 61). Moreover,

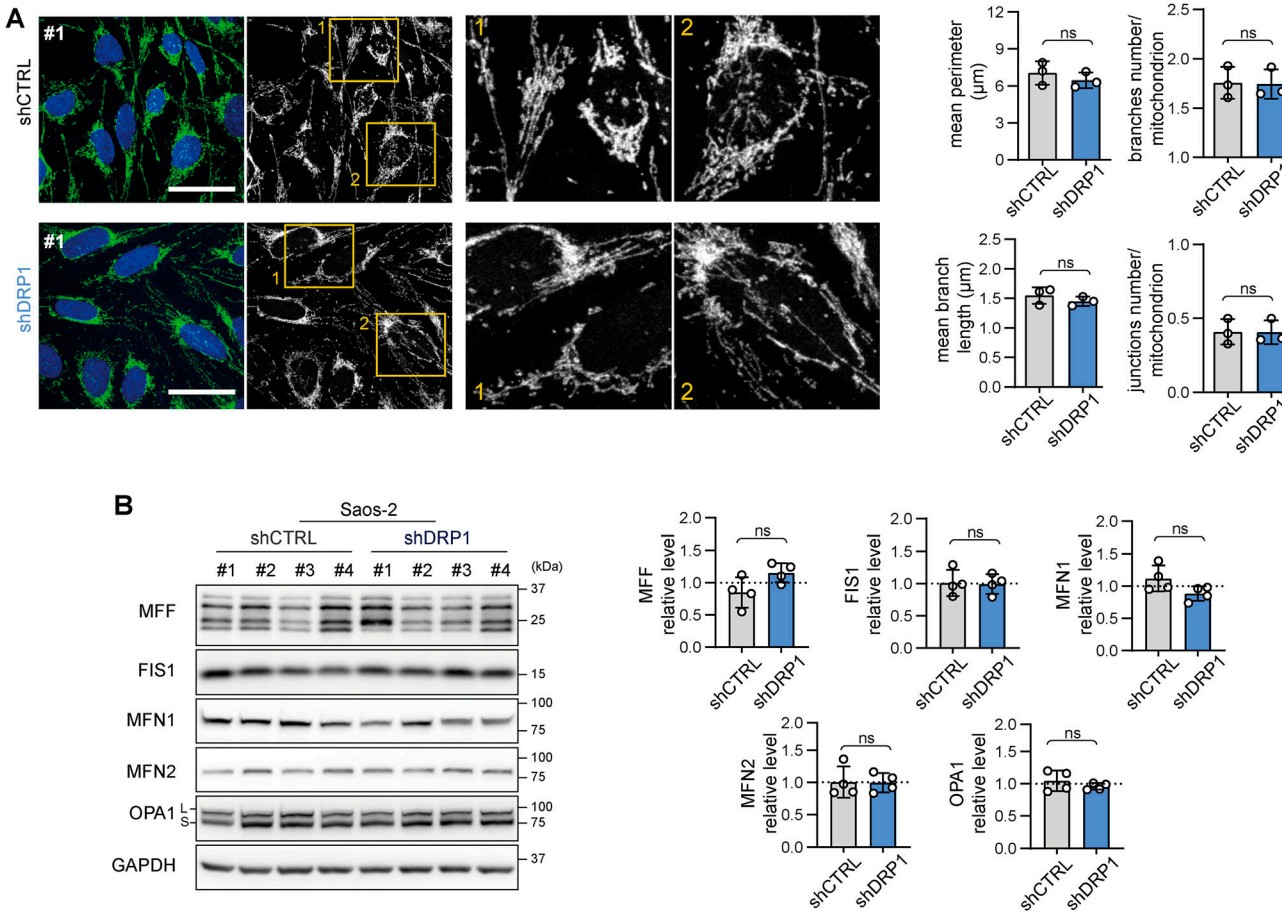

**Figure 7. Depletion of DRP1 does not affect the mitochondrial network morphology in Saos-2 osteosarcoma cells.**
**(A)** Mitochondrial morphology, as visualized by immunofluorescence staining of TOMM20 (green), was not altered by DRP1 down-regulation. Representative merged images (green: TOMM20, blue: nuclei) and grayscale images (mitochondria) of maximum intensity projections of confocal microscopy Z-stacks are shown for individual Saos-2 control (shCTRL) and DRP1–down-regulated (shDRP1) cell clones. White bar: 40 $\mu$m. Images of mitochondria from other analyzed clones are provided in Fig S15. Image analysis using the ImageJ plug-in tool Mitochondria Analyzer showed that parameters describing the mitochondrial network morphology did not significantly differ between the shCTRL and shDRP1 clones. The mean parameters determined for the shCTRL and shDRP1 clone groups are presented as the mean ± SD. Analysis of biological replicates of individual cell clones is provided in Fig S16. **(B)** Western blotting and densitometric analysis of mitochondrial fission– and fusion–related proteins in Saos-2 shCTRL and shDRP1 cell clones. Normalized protein levels detected in the shCTRL and shDRP1 clone groups are plotted relative to the average level as the mean ± SD. Densitometric analysis of biological replicates of individual cell clones is provided in Fig S18. Statistical significance was determined by unpaired two-tailed $t$ test; ns, not significant.
Source data are available for this figure.

similar findings have been observed in other malignancies, such as melanoma (62), lung adenocarcinoma (12), and hepatocellular carcinoma (10), where high expression of mitochondrial fusion–related genes is linked to a poor prognosis.

Although recent studies have demonstrated variable effects of chemotherapy on DRP1 (28, 32, 63) and other proteins involved in mitochondrial fission (24, 61) and fusion (32, 63, 64), the underlying determinants are poorly understood. Using a panel of pediatric sarcoma cell lines with differential DRP1 expression and activation, we demonstrated that the chemotherapy-induced modulation of mitochondrial fission machinery depends on the initial status of DRP1 (Figs 1B and D, 2C and D, and 4A–C). The activity of ERK was previously reported to enhance DRP1-activating phosphorylation (8, 65). However, in RD cells, the increase in ERK activation did not correspond to the chemotherapy-induced activating

phosphorylation of DRP1 (Figs 2C and D and S24). Thus, we hypothesize that other putative mechanisms, such as DUSP6-mediated dephosphorylation of DRP1 at S616, may also contribute to the regulation of DRP1 phosphorylation in rhabdomyosarcoma (66).

Phosphorylation at S616 is widely considered the primary activating modification of DRP1 (7, 40). However, DRP1 activity is further fine-tuned by other posttranslational modifications, including SUMOylation (67), ubiquitylation (68), and phosphorylation at additional residues (69). For example, phosphorylation at S637 has been reported to inhibit DRP1-mediated mitochondrial fission (37, 70), although there are conflicting findings regarding whether this effect is because of impaired recruitment of DRP1 to the mitochondria (71, 72). Future studies investigating how these DRP1 modifications are affected by chemotherapy exposure could

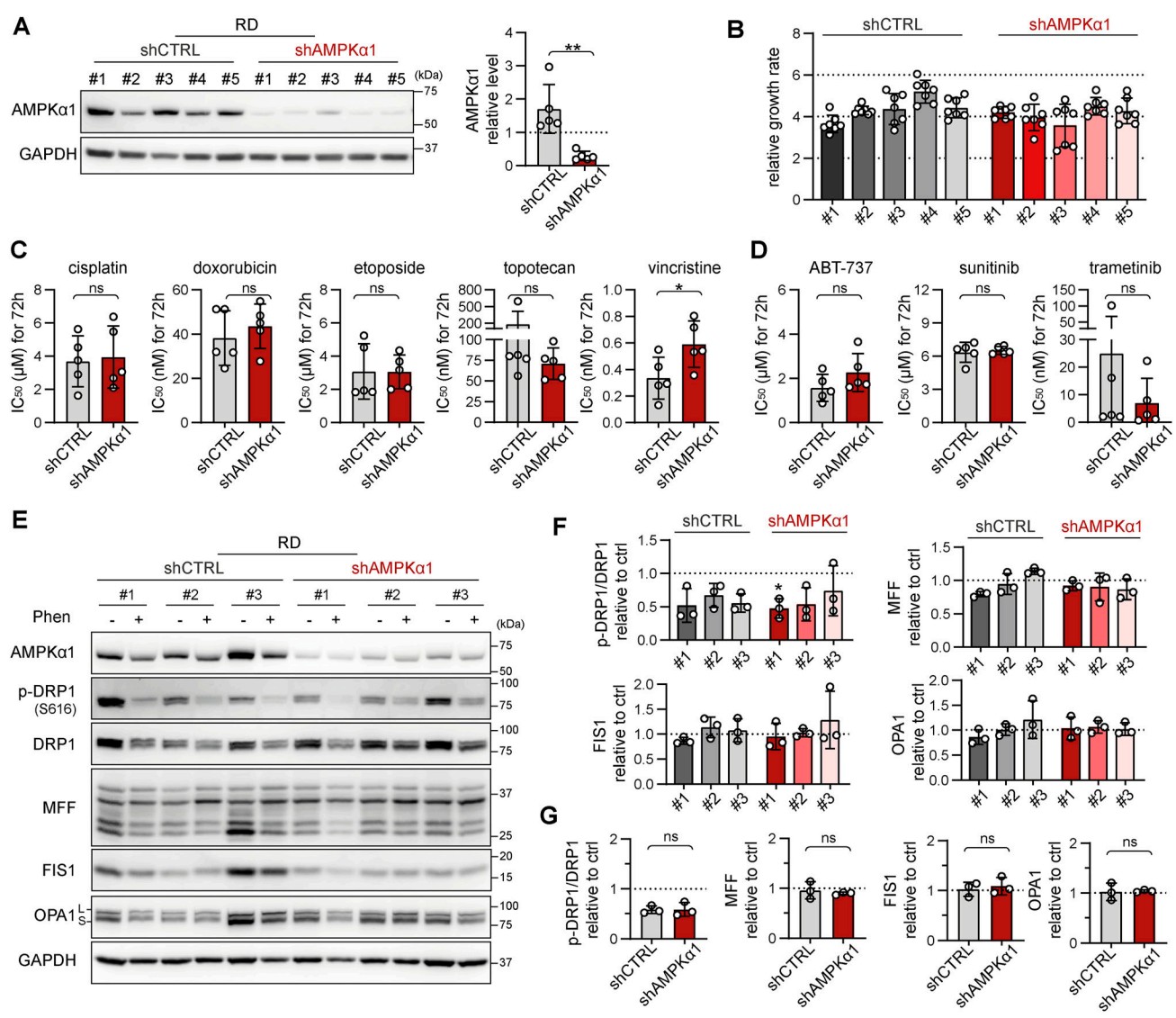

**Figure 8. Rhabdomyosarcoma drug resistance and stress-induced modulation of mitochondrial dynamics–related proteins are independent of the upstream mitochondrial fission and autophagy inducer AMPKα1.**
**(A)** Western blotting and densitometric analysis confirmed the efficiency of AMPKα1 knockdown in rhabdomyosarcoma RD cell clones. The mean normalized protein levels detected in the control (shCTRL) and AMPKα1–down-regulated (shAMPKα1) cell clone groups are plotted relative to the average level as the mean ± SD. Densitometric analysis of biological replicates of individual cell clones is provided in Fig S19A. **(B)** Live-cell imaging analysis of the growth rate of individual shCTRL and shAMPKα1 RD cell clones. The data are plotted as the mean ± SD of the fold change in cell confluence after 72 h of incubation. **(C, D)** The chemoresistance of control and AMPKα1–down-regulated RD cells was compared using the drug IC50. The data are plotted as the mean ± SD of the IC50 values determined for individual shCTRL and shAMPKα1 cell clones. **(C, D)** Dose–response curves for standard chemotherapy drugs (C) and targeted inhibitors (D) are provided in Fig S20. **(E, F, G)** Western blotting (E) and densitometric analysis (F, G) of mitochondrial fission– and fusion–related proteins in rhabdomyosarcoma RD shCTRL and shAMPKα1 cell clones exposed for 72 h to 1 mM phenformin. **(F)** Normalized protein levels detected in individual cell clones are plotted relative to the untreated controls as the mean ± SD (F). **(G)** Comparison of the phenformin-induced modulation of mitochondrial fission– and fusion–related proteins detected in the RD shCTRL and shAMPKα1 cell clone groups (G). Average relative protein levels per group and biological replicate are plotted as the mean ± SD. Densitometric analysis of biological replicates of individual cell clones and extended analysis of fusion-related proteins after 72 h of phenformin treatment are provided in Fig S22. **(C, D, F, G)** Statistical significance was determined by an unpaired two-tailed *t* test (C, D, G) or by one-way ANOVA followed by Tukey's multiple comparisons test (F), *P < 0.05; ns, not significant.
Source data are available for this figure.

provide further insights into the role of mitochondrial fission machinery in tumor cell adaptation to therapy-induced stress.

Although depletion of DRP1 (27, 58, 59) or its adaptor protein MFF (25, 26) has been shown to suppress drug resistance in carcinomas, stable DRP1 knockdown did not mitigate drug resistance in our sarcoma models (Figs 5, S10, and S11). Strikingly, DRP1 depletion

even slightly increased resistance to cisplatin and vincristine in RD cells, likely indicating a predominant proapoptotic role of DRP1 in rhabdomyosarcoma, as found in other cancers (73, 74). Notably, we observed that DRP1 knockdown had a very limited impact on sarcoma cell physiology. In contrast to reports indicating that DRP1 activity is necessary for proper cell cycle progression (75, 76) and

tumor growth (8, 18, 58), the growth rate of sarcoma cells was unaffected by reduced DRP1 levels (Figs 5B and F, S8, and S9). Although previous studies suggested that DRP1 inhibits the cancer stem-like phenotype (18, 77), we did not observe any down-regulation of stemness-associated markers in shDRP1 rhabdomyosarcoma RD cells (Fig S25). Importantly, DRP1 depletion in sarcoma models did not alter the mitochondrial network morphology (Figs 6A and 7A), whereas numerous reports in other cell types have shown that DRP1 knockdown induces a more fused and less fragmented mitochondrial network (27, 58, 60, 75, 78). Taken together, our results suggest the presence of a potential DRP1-independent mechanism that may efficiently compensate for the lack of DRP1 activity in sarcoma cells. Although DRP1 activity is essential for canonical mitochondrial fission (79, 80), dynamin-2 has been reported to be an alternative mediator of mitochondrial fission (81). In addition, the DRP1 adaptor protein FIS1 has been shown to regulate mitochondrial fission by inhibiting mitochondrial fusion (53). Similar compensatory mechanisms might also be adopted by sarcoma cells, as evidenced by the up-regulation of the DRP1 adaptor proteins MFF and FIS1 in DRP1-depleted rhabdomyosarcoma RD cells (Figs 6B and S17), further suggesting that therapies targeting DRP1 for sarcoma may be likely to fail.

Overall, our study revealed that drug exposure modulates mitochondrial fission–related proteins in pediatric sarcomas. Importantly, we revealed a novel finding: the nature of chemotherapy-induced modulation appears to be influenced by the initial status of the DRP1 fission machinery. However, further research is needed to elucidate the possible clinical implications. Strikingly, stable DRP1 knockdown neither mitigated sarcoma chemoresistance nor impacted cell physiology, possibly because of the selection of cells capable of activating compensatory mechanisms. Thus, our results prompt investigations into noncanonical, DRP1-independent mitochondrial fission mechanisms, including upstream regulators, to better understand whether targeting mitochondrial dynamics could serve as a therapeutic strategy to overcome multidrug resistance.

# Materials and Methods

### Cell culture and treatment

The following sarcoma cell lines were used in the study: (i) the osteosarcoma cell line OSA-13 and the embryonal rhabdomyosarcoma cell lines NSTS-11 and NSTS-46, all of which were derived in-house from tumor tissue with written informed consent under the IGA MZCR NR/9125-4 project approved by the Research Ethics Committee of the School of Medicine, Masaryk University, Brno, Czech Republic (approval no. 23/2005); and (ii) the osteosarcoma cell line Saos-2 and the rhabdomyosarcoma cell line RD, both of which were purchased from ECACC. In addition, the human embryonic kidney cell line HEK293T served as a positive control for detecting AMPKα2, as shown in Figs S19, S22, and S23. The authenticity of the cell lines was verified by STR profiling (Generi Biotech), and all cell lines were regularly tested for mycoplasma contamination by PCR (82).

The cell lines were maintained at 37°C in a humidified atmosphere of 5% $CO_2$ in culture media with supplements, as detailed in Table S1. The analyzed cells were treated with drugs (as detailed in Table S2) the day after cell seeding. For treatments involving nutrient deprivation, the cultivation media was replaced immediately before treatment with DMEM without glucose (11966025; Gibco) and without additional supplements, that is, FBS and glutamine.

### Stable knockdown of DRP1 and AMPKα1

The stable knockdown cells and their respective controls were prepared by lentiviral transduction of constructs encoding target-specific shRNA or scramble shRNA under a constitutive promoter (shDRP1: sc-43732-V, shAMPKα1: sc-29673-V, scramble control shCTRL: sc-108080, all from Santa Cruz Biotechnology, Inc.) according to the manufacturer's instructions. Subsequently, efficiently transduced cells were selected by two rounds of 5-d treatment with 2.5 μg/ml puromycin (sc-108071A; Santa Cruz Biotechnology). Individual cell clones were isolated by limiting dilution seeding in 96-well plates and exposed to 2.5 μg/ml puromycin every sixth passage.

### Cell viability MTT assay

Cells were seeded into 96-well plates at a density of 2,000 cells/well. After a 72-h drug treatment incubation period, an MTT assay was performed. Thiazolyl blue tetrazolium bromide (#M2128; Sigma-Aldrich) was added to reach a final concentration of 0.455 mg/ml, and the plates were incubated for 3 h under standard conditions. Subsequently, the medium was aspirated, and the formazan crystals were solubilized with 200 μl of DMSO. The absorbance of each well was measured using a Sunrise Absorbance Reader (Tecan).

To compare drug sensitivity, absolute half-maximal inhibitory concentrations ($IC_{50}$) were determined for individual cell lines and cell clones. They were derived from nonlinear regression of MTT assay datasets, which consisted of absorbance values of individual tested concentrations normalized to those of untreated control cells. Nonlinear regression with variable slope was performed using GraphPad Prism 8.0.2 software (GraphPad Software). Absolute $IC_{50}$ values were calculated from the nonlinear regression parameters using the following formula: relative $IC_{50}*(((50-top)/(bottom-50))^{(-1/hill\ slope)})$.

### Cell cycle analysis

Cells were seeded in a 60-mm Petri dish and harvested using Accutase (#LM-T1735; Biosera) 48 h after seeding and fixed for 30 min with 70% ethanol at 4°C. After fixation, the cells were washed with PBS and stained for 30 min at 37°C with solution containing 0.05 mg/ml propidium iodide (P4170; Sigma-Aldrich), 0.05 mg/ml RNase (D106; Top-Bio), 10 mM Tris, and 10 mM NaCl. Subsequently, fluorescence in the ECD (610/20 nm) channel, excited by a 488-nm laser, was measured using the CytoFLEX S flow cytometer (Beckman Coulter).

## Cell growth analysis

Cells were seeded in at least three technical replicates into 96-well plates at a density of 1,500 cells/well. The day after seeding, cell confluency was determined using an IncuCyte SX1 live-cell imaging system (Sartorius) every 4 h over a 72-h period to monitor cell growth continuity. The cell growth rate was calculated as the ratio of cell confluence in each individual well at the end and beginning of the 72-h analysis.

## Western blotting

Analyzed cells were harvested in RIPA lysis buffer (2 mM EDTA, 1% IGEPAL CA-630, 0.1% SDS, 8.7 mg/ml sodium chloride, 5 mg/ml sodium deoxycholate, 50 mM Tris–HCl) supplemented with cOmplete Mini Protease Inhibitor Cocktail (#11836170001; Roche) and PhosSTOP (#4906837001; Roche) to prepare whole-cell extracts. A total of 20 $\mu$g of protein was denatured using Laemmli sample buffer and heat, loaded into 10% polyacrylamide gels, electrophoretically resolved, and transferred onto PVDF membranes (#1620177; Bio-Rad Laboratories). The membranes were then blocked with a solution of 5% nonfat dry milk or bovine serum albumin (#A7906; Sigma-Aldrich) in Tris-buffered saline supplemented with 0.05% Tween-20 (#93773; Sigma-Aldrich) for at least 1 h, followed by overnight incubation at 4°C with primary antibodies on a rocking platform. Next, the membranes were incubated for at least 1 h at room temperature with secondary HRP-linked antibodies. Details of the antibodies, including dilutions, respective blocking agents, and sample heat denaturation temperatures, are provided in Table S3. Chemiluminescence was detected after a 5-min incubation with ECL Prime Western Blotting Detection Reagent (#RPN2236; Cytiva) using the Azure C600 imaging system (Azure Biosystems).

Densitometric image analysis was performed using the gel analysis tool in ImageJ (Fiji) software (NIH), version 2.1.0/1.53c. The detected signal of the protein of interest was normalized to that of the loading control (GAPDH, $\alpha$-tubulin, or $\beta$-actin) provided on the same gel. The levels of phosphorylated proteins (p-DRP1[S616], p-AMPK$\alpha$[Thr172], p-ERK1/2[T202/YT204]) were analyzed as the ratio of the normalized phosphorylated form to the normalized total protein form (DRP1, AMPK$\alpha$1, and ERK1/2, respectively). Original uncropped blots of all replicates are provided as Source Data files.

## Immunostaining

Cells were seeded on glass coverslips. After 2 d of cultivation, the cells were rinsed with PBS, fixed with 3% paraformaldehyde (#158127; Sigma-Aldrich) for 20 min, and permeabilized with 0.2% Triton X-100 (#04807423; MP Biomedicals) for 1 min. The fixed and permeabilized cells were then exposed to a blocking solution (3% BSA [#A7906; Sigma-Aldrich] in PBS) for 20 min and incubated with primary and subsequently secondary antibodies diluted in the blocking solution for at least 1 h at 37°C. The list of primary and secondary antibodies is provided in Table S3. Nuclei were stained with 1 $\mu$g/ml Hoechst 33342 (#H1399; Invitrogen). The coverslips were mounted using ProLong Diamond Antifade (#P36961; Invitrogen). Z-stacked images were captured using a Leica SP8 confocal

microscope (Leica) and processed as maximum intensity projections using LAS X software (Leica, 3.4.218368).

## Mitochondrial network morphology analysis

The ImageJ (Fiji) plug-in tool Mitochondrial Analyzer (83) was used to analyze the mitochondrial network morphology on a per-field-of-view basis from 2D maximum intensity projections of z-stack confocal images of the translocase of outer mitochondrial membrane 20 fluorescence channel. The preprocessing parameters were set as follows: rolling (microns), 1; radius, 2; max slope, 2; and gamma, 0.8. The applied commands included subtracting the background, Sigma-Aldrich filter plus, enhancing local contrast, and adjusting gamma. The threshold method used was the weighted mean method with a block size of 1.75 $\mu$m and a c value of 1.25. Postprocessing commands included despeckling and removing outliers with a 2-pixel radius. At least 2000 mitochondria in at least four fields of view were analyzed for each biological replicate.

## Gene expression transcriptomic analysis

The R2: Genomics Analysis and Visualization Platform (http://r2.amc.nl) was used for single-gene survival analysis. The expression of the mitochondrial fission mediator DRP1-encoding gene *DNM1L1* was used to group patients, for whom Kaplan–Meier survival curves were generated from publicly available osteosarcoma (Kuijjer) (39) and rhabdomyosarcoma (Williamson) (38) transcriptomic datasets. Clinical characteristics of the patients are detailed in the respective publications (38, 39). Kaplan–Meier curves were plotted for the high (blue) and low (red) expression groups established by the scan cutoff mode with a minimal group size of 20 patients. In addition, the *P*-value was calculated from the plotted curves using a two-sided log-rank test.

## Statistical analysis

All experiments were performed with at least three independent biological replicates; further details are provided in the figure legends. Line graphs and bar graphs display the mean ± SD. In the bar graphs, individual data points represent independent biological replicates, except for the general comparison of the control and knockdown cell clone groups in Figs 5A and E, 6, 7, 8A and G, S8B, S9B, S22C, S23D, and S25B, where individual data points refer to the mean value determined for individual cell clones in at least three independent biological replicates. Further exceptions are Figs 5C, D, F, and G, 8C and D, S12, and S21, where individual data points represent the IC50 determined for each individual cell clone. Statistical analysis of Kaplan–Meier survival curves was performed using the R2: Genomics Analysis and Visualization Platform (http://r2.amc.nl) with a two-sided log-rank test; otherwise, statistical analysis was conducted using GraphPad Prism 8.0.2. software. An unpaired two-tailed *t* test was applied when comparing two groups; otherwise, one-way ANOVA followed by Tukey's multiple comparison test was used, assuming a normal data distribution and similar variance between the compared groups. *P*-values < 0.05 were considered to indicate statistical significance; *$P < 0.05$, **$P < 0.01$, ***$P < 0.001$, #$P < 0.0001$; ns refers to a *P*-value ≥ 0.05.

## Data Availability

All data are included in the article and supporting information. Additional supporting data are available from the corresponding author upon reasonable request.

## Supplementary Information

## Acknowledgements

This work was supported by the Czech Science Foundation (No. GJ20-00987Y). J Skoda and K Borankova acknowledge the project National Institute for Cancer Research (Programme EXCELES, ID Project No. LX22NPO5102)—Funded by the European Union—Next Generation EU. The authors also express their gratitude to Johana Marešová and Dagmar Štodtová for their skilled technical assistance.

### Author Contributions

K Borankova: conceptualization, data curation, formal analysis, supervision, investigation, visualization, methodology, and writing—original draft, review, and editing.
M Solny: data curation, formal analysis, investigation, visualization, and writing—review and editing.
M Krchniakova: formal analysis, investigation, and writing—review and editing.
J Skoda: conceptualization, resources, supervision, funding acquisition, visualization, methodology, project administration, and writing—original draft, review, and editing.

### Conflict of Interest Statement

The authors declare that they have no conflict of interest.

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
