## [Reviewer comments · Life Science Alliance]

Life Science Alliance

Depleting chemoresponsive mitochondrial fission mediator DRP1 does not mitigate sarcoma resistance

Karolina Borankova, Matyas Solny, Maria Krchniakova, and Jan Skoda

DOI: [10.26508/lsa.202402870](https://doi.org/10.26508/lsa.202402870)

Corresponding author(s): Jan Skoda, Masaryk University

Review Timeline:

Submission Date:	2024-06-05
Editorial Decision:	2024-08-12
Revision Received:	2024-11-10
Editorial Decision:	2024-11-25
Revision Received:	2024-11-28
Accepted:	2024-11-29

Transaction Report:

August 12, 2024

Re: Life Science Alliance manuscript #LSA-2024-02870

Jan Skoda

Dear Dr. Skoda,

Thank you for submitting your manuscript entitled "Depletion of chemoresponsive mitochondrial fission mediator DRP1 does not mitigate sarcoma resistance" to Life Science Alliance. The manuscript was assessed by expert reviewers, whose comments are appended to this letter. We invite you to submit a revised manuscript addressing the Reviewer comments.

Thank you for this interesting contribution to Life Science Alliance. We are looking forward to receiving your revised manuscript.

Sincerely,

B. MANUSCRIPT ORGANIZATION AND FORMATTING:

Reviewer #1 (Comments to the Authors (Required)):

Summary

The manuscript investigates the role of mitochondrial dynamics in pediatric sarcomas, specifically rhabdomyosarcoma and osteosarcoma. It focuses on the modulation of mitochondrial fission and fusion-related proteins following chemotherapy exposure and explores the potential of targeting these dynamics to overcome drug resistance.

Key Findings

- 1) Chemotherapy-induced modulation of DRP1: The study shows that chemotherapy affects the mitochondrial fission protein DRP1 differently in rhabdomyosarcoma and osteosarcoma. High DRP1 levels are associated with better outcomes in rhabdomyosarcoma (RD) but worse outcomes in osteosarcoma based on patient survival data.
- 2) Genetic manipulation of DRP1: Despite the observed correlations, genetic manipulation of DRP1 does not significantly change cell growth or chemoresponse in vitro. This suggests that other factors might be compensating for the loss of DRP1 function.
- 3) Potential regulators of DRP1: The manuscript explores potential regulators of DRP1, such as mitophagy, AMPK, and ERK, to understand their roles in modulating DRP1 activity and their implications for sarcoma cell survival and drug resistance.

Strengths

- 1) Relevance to pediatric oncology: The study addresses the critical issue of multidrug resistance in pediatric sarcomas, a significant challenge in improving patient outcomes. By focusing on mitochondrial dynamics, the research explores a relatively under-investigated area that could offer new therapeutic targets.
- 2) Detailed literature review: The introduction provides a thorough review of existing literature on mitochondrial dynamics, highlighting the knowledge gap in pediatric sarcomas and setting a solid foundation for the study's objectives.
- 3) Clear research objective: The manuscript clearly states its aim to analyze the modulation of mitochondrial fission and fusion-related proteins following drug exposure in pediatric sarcomas, providing a focused direction for the study.
- 4) Clinical outcomes of DRP1 levels: The study analyzes patient data to show different clinical outcomes based on DRP1 levels. It clearly demonstrates that chemotherapy induces DRP1 levels, suggesting a potential role in overcoming chemotherapy resistance. It also explores potential modulators regulating DRP1 levels and activation, such as autophagy and AMPK.
- 5) Comprehensive discussion: The discussion integrates the study findings with existing literature, providing novel insights into mitochondrial dynamics in pediatric sarcomas. It explores potential clinical implications and suggests further research directions, demonstrating the study's relevance and impact.

Weaknesses and Recommendations

- 1) In Figure 2C, the LC3AB bands in the control (-/-) and BAF only conditions show unexpected variations across different drug combination panels (CIS, DXR, TOP, VIN). These should be relatively similar, given that these groups represent baseline conditions. This raises questions about the reproducibility and reliability of autophagy flux measurements.
- 2) Given the data in Figures 6 and 7 suggesting that alterations in DRP1 do not change mitochondrial morphology, the statement in line 109, "This finding suggested that activation of DRP1-mediated mitochondrial fission serves as an adaptation to chemotherapy-induced stress," is overstated because the authors did not show direct measurements of mitochondrial morphology (fission) in cells following chemotherapy. Additional immunofluorescence imaging with TOM20 in cells following chemotherapy will be necessary for this statement.
- 3) The statement in line 162 "Despite significantly decreased DRP1 activity, the growth rates of rhabdomyosarcoma (Fig. 5b) and osteosarcoma (Fig. 5f) shDRP1 clones were very similar to those of their control counterparts, suggesting that DRP1 is not required for sarcoma cell cycle progression" is too strong. The authors only measured cell confluency, which does not reflect potential changes in cell morphology and size that could affect growth measurements. To provide more robust evidence on the impact of DRP1 on cell cycle progression, the authors can perform additional experiments such as flow cytometry or western blotting for cell cycle regulators.

Minor Comments

- 1) Define all abbreviations at their first occurrence to ensure clarity.
- 2) Standardize the terminology for mitochondrial dynamics-related proteins throughout the manuscript. For example, in Figures 6b and 7b, the y-axis labels are not consistent. Choose either the full gene name like Mitofusin-1 and Mitofusin-2 or the abbreviated gene names and use them consistently.

Conclusion

The manuscript provides significant contributions to understanding mitochondrial dynamics in pediatric sarcomas and their potential as therapeutic targets to overcome drug resistance. With improvements in experimental design, data consistency, and the inclusion of functional assays, the study could offer even more robust and comprehensive insights into the role of mitochondrial dynamics in pediatric sarcoma therapy.

Reviewer #2 (Comments to the Authors (Required)):

Mitochondrial dynamics, encompassing fusion and fission processes, are crucial for cellular functions such as energy production, mitophagy, and autophagy. Recent studies have implicated mitochondrial dynamics in contributing to drug resistance in cancer. In this research, the authors utilized multiple sarcoma cell lines to investigate the relationship between various mitochondrial dynamic proteins and drug resistance.

The subject matter of this study is intriguing; however, the quality of the research is suboptimal. The experimental design lacks rigor, data interpretation is unconvincing, and the logical framework of the paper is weak.

Several major concerns need to be addressed:

1. The authors employed several sarcoma cell lines to assess the impact of mitochondrial division mechanisms on drug resistance. However, discrepancies among these cell lines were evident in most of the presented data, undermining the robustness of the conclusions drawn. Furthermore, essential controls were omitted in the study.
2. The authors explored the effects of Drp1 phosphorylation, focusing on the serine 616 site and assuming it to be an "activating site." However, the field remains divided on the role of phosphorylation. Clarifying the effects of S616 on Drp1 morphology and utilizing phosphomimic mutants with high-quality imaging would strengthen their claims.
3. The first four figures suggest Drp1's involvement in sarcoma drug resistance, whereas later figures suggest it may not be essential, creating confusion. The manuscript requires better organization and explanation to maintain coherence throughout.

Minor concerns include:

1. Figure 1a lacks important Drp1 regulators such as MiD49 and MiD51, known Drp1 receptors. Additionally, several publications in the field have shown that FIS1 is not Drp1 receptor in mammalian cells. Furthermore, Drp1 activity is not solely governed by the S616 site; phosphorylation at S637 is also critical for mitochondrial fission. References to recent studies demonstrating the interplay between these phosphorylation sites should be included.
2. In Figure 1b, the use of transcriptomic datasets to correlate Drp1 expression levels with sarcoma lacks a rigorous experimental description. Patient ages, demographics and clinical conditions are crucial for interpreting such data. The criteria for defining high/low expression levels should also be clearly specified.
3. The experimental design for Figures 1c and 1d raises concerns; comparing different cell lines instead of pre- and post-therapy within the same cell line might lead to misleading conclusions due to variations in mitochondrial division machinery profiles among different cells.
4. The claim that Drp1 S616 was regulated in 2 out of 3 post-therapy cell lines suggests the mechanism may not be universally applicable. Caution is advised in drawing conclusions, particularly in Figure 1 where the title asserts enhanced Drp1 activation post-therapy.
5. Figures 2c and 2d suffer from similar issues; the observed increase in Drp1 S616 levels under chemotherapy is inconsistent across different drug groups. The increasing is not detected in the VIN group. Moreover, total Drp1 levels increase in the TOP group, affecting the applicability of the conclusions drawn.
6. Figure 3 lacks statistical analysis comparing the curves with and without rapamycin treatment, which is necessary for robust interpretation.
7. In Figure 6, the authors conclude that depletion of Drp1 does not affect mitochondrial network morphology; however, the quality of the images is inadequate for reliable quantification, zoom field analysis is recommended. Additionally, the observed clustering of mitochondria in perinuclear regions in Drp1 KD cells is atypical and suggests potential issues with cell health.

Rebuttal Letter – LSA-2024-02870: “Depletion of chemoresponsive mitochondrial fission mediator DRP1 does not mitigate sarcoma resistance”

We would like to cordially thank the reviewers for their detailed analysis of our manuscript and for suggestions to improve it. We also greatly appreciate the summary of the manuscript's strengths and weaknesses. We carefully considered the recommendations and have addressed the outstanding questions by performing new analyses and revising the manuscript, as detailed in the point-by-point response below.

Reviewer #1**Summary**

The manuscript investigates the role of mitochondrial dynamics in pediatric sarcomas, specifically rhabdomyosarcoma and osteosarcoma. It focuses on the modulation of mitochondrial fission and fusion-related proteins following chemotherapy exposure and explores the potential of targeting these dynamics to overcome drug resistance.

Key Findings

- 1) Chemotherapy-induced modulation of DRP1: The study shows that chemotherapy affects the mitochondrial fission protein DRP1 differently in rhabdomyosarcoma and osteosarcoma. High DRP1 levels are associated with better outcomes in rhabdomyosarcoma (RD) but worse outcomes in osteosarcoma based on patient survival data.*
- 2) Genetic manipulation of DRP1: Despite the observed correlations, genetic manipulation of DRP1 does not significantly change cell growth or chemoresponse in vitro. This suggests that other factors might be compensating for the loss of DRP1 function.*
- 3) Potential regulators of DRP1: The manuscript explores potential regulators of DRP1, such as mitophagy, AMPK, and ERK, to understand their roles in modulating DRP1 activity and their implications for sarcoma cell survival and drug resistance.*

Strengths

- 1) Relevance to pediatric oncology: The study addresses the critical issue of multidrug resistance in pediatric sarcomas, a significant challenge in improving patient outcomes. By focusing on mitochondrial dynamics, the research explores a relatively under-investigated area that could offer new therapeutic targets.*
- 2) Detailed literature review: The introduction provides a thorough review of existing literature on mitochondrial dynamics, highlighting the knowledge gap in pediatric sarcomas and setting a solid foundation for the study's objectives.*
- 3) Clear research objective: The manuscript clearly states its aim to analyze the modulation of mitochondrial fission and fusion-related proteins following drug exposure in pediatric sarcomas, providing a focused direction for the study.*
- 4) Clinical outcomes of DRP1 levels: The study analyzes patient data to show different clinical outcomes based on DRP1 levels. It clearly demonstrates that chemotherapy induces DRP1 levels, suggesting a potential role in overcoming chemotherapy resistance. It also explores potential modulators regulating DRP1 levels and activation, such as autophagy and AMPK.*
- 5) Comprehensive discussion: The discussion integrates the study findings with existing literature, providing novel insights into mitochondrial dynamics in pediatric sarcomas. It explores potential clinical implications and suggests further research directions, demonstrating the study's relevance and impact.*

Response: We are thankful for the reviewer's thoughtful and encouraging feedback, recognizing the overall quality of our work and its relevance to the field.

Weaknesses and Recommendations

1) In Figure 2C, the LC3AB bands in the control (-/-) and BAF only conditions show unexpected variations across different drug combination panels (CIS, DXR, TOP, VIN). These should be relatively similar, given that these groups represent baseline conditions. This raises questions about the reproducibility and reliability of autophagy flux measurements.

Response: We agree with the reviewer that the blots presented in original Fig. 2c suggested the unexpected variation of the autophagy flux in untreated controls (-/-) and BAF-only) among the different treatment groups. However, we are convinced this rather reflected technical variation (different intensity of protein bands; inevitable in different western blotting runs) and inappropriate selection of representative blots. As apparent from Fig. 2d, treatment with BAF alone resulted in similar accumulation of LC3AB-II across the treatment groups, confirming the biological reproducibility of the basal autophagy flux. Despite potential technical variability, when the BAF-only values are normalized to respective controls (-/-) on the same blot, they are similar across the treatment groups. We now also provide the Source Data file for Fig. 2c, including all biological replicates, which support the reliability of the determined changes in autophagy flux upon chemotherapy treatment. To avoid any confusion, we have selected different blots (i.e., different biological replicates) for the revised version of Fig. 2c, which better represents all biological replicates for the CIS, DXR, and TOP drug combinations.

2) Given the data in Figures 6 and 7 suggesting that alterations in DRP1 do not change mitochondrial morphology, the statement in line 109, "This finding suggested that activation of DRP1-mediated mitochondrial fission serves as an adaptation to chemotherapy-induced stress," is overstated because the authors did not show direct measurements of mitochondrial morphology (fission) in cells following chemotherapy. Additional immunofluorescence imaging with TOM20 in cells following chemotherapy will be necessary for this statement.

Response: We appreciate this important observation and agree with the reviewer that the statement made in line 109 was not sufficiently supported by the current experimental data. To ensure accurate data interpretation, we have revised the statement accordingly.

Revised manuscript lines 110-112: 'This finding suggested that the increase in DRP1 activating phosphorylation at S616 might serve as an adaptation to chemotherapy-induced stress and that inhibiting autophagy, thereby mitophagy, prevents this response.'

3) The statement in line 162 "Despite significantly decreased DRP1 activity, the growth rates of rhabdomyosarcoma (Fig. 5b) and osteosarcoma (Fig. 5f) shDRP1 clones were very similar to those of their control counterparts, suggesting that DRP1 is not required for sarcoma cell cycle progression" is too strong. The authors only measured cell confluency, which does not reflect potential changes in cell morphology and size that could affect growth measurements. To provide more robust evidence on the impact of DRP1 on cell cycle progression, the authors can perform additional experiments such as flow cytometry or western blotting for cell cycle regulators.

Response: We thank the reviewer for this insightful and constructive feedback. We agree that cell confluency measurements are insufficient to draw conclusions regarding cell cycle progression. In response, we performed cell cycle analysis by flow cytometry in the individual control (shCTRL) and DRP1-downregulated (shDRP1) clones. These results are now provided in Fig 5B,F, Fig S8 and Fig S9. Consistent with the similar growth rates observed previously (data now moved into Fig S8 and Fig S9), we did not detect any significant differences in the G0/G1, S, or G2/M phases between the sarcoma shCTRL and shDRP1 groups. We believe this additional experimental analysis strengthens our conclusion regarding the role of DRP1 in sarcoma cell cycle progression.

Revised manuscript lines 166-170: 'Despite significantly decreased DRP1 activating phosphorylation, the cell cycle progression and growth rates of rhabdomyosarcoma (Fig 5B, Fig S8) and osteosarcoma shDRP1 clones (Fig 5F, Fig S9) were very similar to those of their control counterparts, suggesting that DRP1 is not required for sarcoma cell proliferation.'

Minor Comments

1) Define all abbreviations at their first occurrence to ensure clarity.

Response: We appreciate this reviewer's observation. We carefully revised the manuscript and added the missing definitions.

Revised manuscript lines 63-64: Notably, we showed that the expression and activating phosphorylation of the mitochondrial fission mediator dynamin-related protein 1 (DRP1) are modulated in sarcoma cells upon chemotherapy exposure.

Revised manuscript lines 88-91: Interestingly, post-therapy osteosarcoma Saos-2 cells showed upregulated levels of both mitochondrial fission- and fusion-related proteins despite maintaining relatively low mitochondrial mass, as assessed by the expression of the translocase of outer mitochondrial membrane 20 (TOMM20), which serves as a proxy for mitochondrial content.

Revised manuscript lines 116-119: Considering that both cisplatin and doxorubicin enhanced autophagic flux in RD cells, as indicated by the increase in the isoform II of microtubule-associated protein 1 light chain 3A and 3B (LC3AB-II) upon bafilomycin A1-mediated inhibition of autophagy (Fig. 2c, d), we speculated that, in line with reports in other cell types (33,46,47), autophagy might protect rhabdomyosarcoma cells against these genotoxic drugs.

Revised manuscript lines 139-141: A decrease in total DRP1 was detected in NSTS-11 cells, while NSTS-46 and OSA-13 cells—expressing the highest basal levels of the DRP1 adaptor, mitochondrial fission factor (MFF) (Fig. 1c, d)—showed prevalent downregulation of MFF in response to the tested drugs (Fig. 4a-c).

Revised manuscript lines 142-144: Interestingly, the levels of mitochondrial fission 1 protein (FIS1), another DRP1 adaptor protein, did not show any clear trend upon drug exposure (Fig S5), suggesting that only a subset of mitochondrial fission-related proteins are affected by drug treatment.

Revised manuscript lines 148-152: In addition to mitofusin-2 (MFN2), the mediator of inner mitochondrial membrane fusion, mitochondrial dynamin-like GTPase OPA1 (OPA1), remained intact in drug-exposed sarcoma cells (Fig 4A and B, Fig S5), demonstrating that chemotherapy did not activate the mitochondrial stress sensors OMA1 zinc metallopeptidase (OMA1) and ATP-dependent zinc metalloprotease YME1L1 (YME1L1), which cleave OPA1 into its short forms (S-OPA1) under mitochondrial stress.

2) Standardize the terminology for mitochondrial dynamics-related proteins throughout the manuscript. For example, in Figures 6b and 7b, the y-axis labels are not consistent. Choose either the full gene name like Mitofusin-1 and Mitofusin-2 or the abbreviated gene names and use them consistently.

Response: We thank the reviewer for pointing this out to us. We have revised all figures, ensuring that the abbreviated names are used throughout the manuscript.

Conclusion

The manuscript provides significant contributions to understanding mitochondrial dynamics in pediatric sarcomas and their potential as therapeutic targets to overcome drug resistance. With improvements in experimental design, data consistency, and the inclusion of functional assays, the study could offer even more robust and comprehensive insights into the role of mitochondrial dynamics in pediatric sarcoma therapy.

Response: We are thankful for the reviewer's overall positive feedback. We hope that the revisions made effectively address the concerns raised, and believe they enhanced the clarity and impact of our findings.

Reviewer #2

Mitochondrial dynamics, encompassing fusion and fission processes, are crucial for cellular functions such as energy production, mitophagy, and autophagy. Recent studies have implicated mitochondrial dynamics in contributing to drug resistance in cancer. In this research, the authors utilized multiple sarcoma cell lines to investigate the relationship between various mitochondrial dynamic proteins and drug resistance.

The subject matter of this study is intriguing; however, the quality of the research is suboptimal. The experimental design lacks rigor, data interpretation is unconvincing, and the logical framework of the paper is weak.

Response: We are thankful for the reviewer's thoughtful feedback. We are pleased that they found the subject of our study intriguing, and we hope that the improvements in data analysis and interpretation in this revised manuscript, made in accordance with the provided comments, will address their concerns and enhance the overall quality of the presented study. We understand that there may be differing opinions on the same subject; however, we would like to refer to the report of another reviewer, who highlighted several strengths of our manuscript, including a clear research objective (logical framework) and convincing experimental evidence supporting the key conclusions.

Several major concerns need to be addressed:

1. The authors employed several sarcoma cell lines to assess the impact of mitochondrial division mechanisms on drug resistance. However, discrepancies among these cell lines were evident in most of the presented data, undermining the robustness of the conclusions drawn. Furthermore, essential controls were omitted in the study.

Response: We appreciate the opportunity to address this feedback, although it remains unclear which specific discrepancies the reviewer is referring to. Indeed, we employed several sarcoma cell lines to account for inherent variability in genetic and epigenetic backgrounds as well as differences in acquired therapy resistance (therapy-naïve vs. post-therapy cell lines). As further reiterated in responses below, this allowed us to identify changes shared among the clinically relevant models of therapy-naïve vs. chemoresistant tumor cells. Results in this panel of cell lines pointed to the differences in levels of DRP1, and namely its phosphorylated form p-DRP1(S616) which was markedly increased only in cell lines from therapy-resistant tumors. The biological relevance of these findings was validated by the following experiments, which identified p-DRP1(S616) and/or DRP1 to be generally responsive to chemotherapy treatment.

We believe that the minor discrepancies observed among the different sarcoma cell lines, such as the variable modulations of MFN1 upon topotecan exposure (Fig 2C, Fig 4, Fig S2), do not conflict with our key conclusions regarding the chemotherapy-induced modulation of DRP1. Furthermore, in the light of other reviewer's report, we are convinced that an appropriate experimental design was applied throughout the study and we are not aware of any missing essential controls. Unfortunately, the lack of details in this comment makes it difficult for us to provide a more comprehensive response. We hope that the discrepancies mentioned by the reviewer are specified in the following comments, which we address below.

2. The authors explored the effects of Drp1 phosphorylation, focusing on the serine 616 site and assuming it to be an "activating site." However, the field remains divided on the role of phosphorylation. Clarifying the effects of S616 on Drp1 morphology and utilizing phosphomimic mutants with high-quality imaging would strengthen their claims.

Response: We respectfully disagree with the assertion that the field remains divided on the role of DRP1 S616 phosphorylation. To our knowledge, there is a consensus regarding the activating role of this phosphorylation site. To support this, we refer to recent reviews on the mitochondrial fission machinery – Kraus et al., *Nature* 2021 (PMID: 33536648), Chen et al., *Signal Transduct Target Ther.* 2023 (PMID: 37669960) – and selected recent research articles which repeatedly demonstrate the association of

DRP1 phosphorylation at S616 with enhanced mitochondrial fission – Lai et al., *J Cell Sci.* 2023 (PMID: 37232206), Xiong et al., *Cell Death Differ.* 2022 (PMID: 35332310), Gao et al., *Signal Transduct Target Ther.* 2022 (PMID: 35422062), Ma et al., *Sci Adv.* 2020 (PMID: 32232156), Kashatus et al., *Mol Cell.* 2015 (PMID: 25658205). Indeed, this understanding is also well reflected by papers in high-profile journals, e.g., see Fonseca et al., *Nature* 2019 (PMID: 31217603) stating: “The phosphorylation of DRP1 residues Ser616 (which promotes mitochondrial localization¹⁰) and Ser637 (which excludes DRP1 from mitochondria²)...”. In contrast, a seminal paper from Adam Frost’s group provided structural evidence for the well-established inhibitory effects of S637 phosphorylation, showing that the phosphomimic mutant at DRP1 residues equivalent to S637 failed to bind to the mitochondrial fission adaptor protein MID49 (Kalia et al., *Nature.* 2018; PMID: 29899447).

We acknowledge the recent study by Liu et al. (*Mol Biol Cell* 2024, PMID: 38019609), which indicates that phosphorylation of DRP1 at S616 alone may not be sufficient to enhance its GTPase activity. However, as the authors correctly discussed, this study has several limitations that might explain its contradictory results: (i) DRP1 isoform 3 was used to generate the phosphomimic Drp1 mutant S579D corresponding to S616D in isoform 1; thus, observations may be limited to the isoform 3; (ii) experiments were performed in a cell-free system using a bacterially-expressed, purified DRP1 phosphomimic mutant, which limited the ability to unbiasedly assess interactions with other partners and evaluate direct impact on mitochondrial fission. In agreement with the authors’ discussion, this study does not contradict the characterization of S616 as an activating site but rather suggests that additional components likely play a role in the mitochondrial fission processivity.

While we appreciate the reviewer’s suggestion to utilize phosphomimic mutants, we are convinced this is beyond the scope of the current study. As pointed above, phosphorylation at S616 is well-documented to be stimulatory to DRP1-mediated mitochondrial fission. Moreover, additional validation of the role of S616 phosphorylation would not challenge the key findings of our study, which demonstrated that DRP1 knockdown does not mitigate chemoresistance, making DRP1 a poor therapeutic target in pediatric sarcomas and suggesting that an alternative, DRP1-independent mitochondrial fission mechanism can compensate for the lack of this canonical mitochondrial fission protein.

3. The first four figures suggest Drp1’s involvement in sarcoma drug resistance, whereas later figures suggest it may not be essential, creating confusion. The manuscript requires better organization and explanation to maintain coherence throughout.

Response: We thank the reviewer for this feedback. Indeed, our initial hypothesis, based on the results presented in Fig 1-4, was that DRP1 plays a direct role in sarcoma chemotherapy resistance. We tested this hypothesis using shRNA-mediated knockdown of DRP1. However, our analysis of chemoresistance in control and DRP1-knockdown clones rejected the direct involvement of DRP1 in mediating the therapy resistance. These results mechanistically demonstrate that the differences in the levels of total DRP1 and its phospho-S616 form rather accompany than directly mediate the response to chemotherapeutics. While these changes appear to mark a chemoresistant state, DRP1 knockdown experiments suggested that strategies targeting DRP1 in pediatric sarcomas are unlikely to induce therapeutic response. These key findings are reflected in the manuscript’s title and clearly summarized in its abstract. To better convey our rationale and guide readers through the study results, we have revised the text in the Results section introducing the results from the DRP1 knockdown experiments.

Revised manuscript lines 158-162: ‘Since sarcoma survival was stratified by DRP1 expression (Fig 1B) and upregulated DRP1-activating phosphorylation at S616 was associated with the post-therapy state (Fig 1C and D), with exposure to chemotherapeutics further increasing this phosphorylation (Fig 2C and D, Fig 4A-C), we hypothesized that DRP1 activity is involved in the mechanism that protects sarcoma cells against chemotherapy-induced stress.’

Minor concerns include:

1. Figure 1a lacks important Drp1 regulators such as MiD49 and MiD51, known Drp1 receptors. Additionally, several publications in the field have shown that FIS1 is not Drp1 receptor in mammalian

cells. Furthermore, Drp1 activity is not solely governed by the S616 site; phosphorylation at S637 is also critical for mitochondrial fission. References to recent studies demonstrating the interplay between these phosphorylation sites should be included.

Response: We appreciate the reviewer's input. The intent of Fig 1A was to schematically depict only the proteins analyzed in our study. For clarity, we have specified this in the revised legend of Fig 1A. For a detailed description of the mitochondrial fission and fusion machinery, we refer readers to the recent expert review by Quintana-Cabrera and Scorrano in *Molecular Cell* (2023).

Revised Fig 1A legend: '(A) An overview of mitochondrial fission and fusion-related proteins analyzed in the study. The mitochondrial fission mediator DRP1, which is activated by the phosphorylation of S616, forms a contractile ring on the mitochondrial surface by interacting with adaptor proteins such as mitochondrial fission factor (MFF) and FIS1, leading to mitochondrial constriction and fission. Mitofusin-1 (MFN1) and mitofusin-2 (MFN2) mediate fusion of the outer mitochondrial membrane, while OPA1 mitochondrial dynamin like GTPase (OPA1) mediates fusion of the inner mitochondrial membrane. For further details, including additional components of the mitochondrial fission and fusion machinery, please refer to Ref. (68).'

However, we agree with the reviewer on the importance of addressing additional post-translational modifications of DRP1 beyond phosphorylation at serine 616. Accordingly, we have expanded the discussion in the revised manuscript to include other relevant DRP1 modifications, such as phosphorylation at serine 637.

Revised manuscript lines (241-247): Phosphorylation at S616 is widely considered the primary activating modification of DRP1 (7,37). However, DRP1 activity is further fine-tuned by other post-translational modifications, including SUMOylation (64), ubiquitylation (65), and phosphorylation at additional residues (66). For example, phosphorylation at S637 has been reported to inhibit DRP1-mediated mitochondrial fission (67,68), although there are conflicting findings regarding whether this effect is due to impaired recruitment of DRP1 to the mitochondria (69,70). Future studies investigating how these modifications are affected by chemotherapy exposure could provide further insights into the role of mitochondrial fission machinery in tumor cell adaptation to therapy-induced stress.

In contrast, we respectfully disagree with the reviewer's statement suggesting that FIS1 is not a DRP1 receptor in mammalian cells. While Otera et al. (*J Cell Biol.* 2010 PMID: 21149567) reported ongoing mitochondrial fission in FIS1-depleted cells, to the best of our knowledge, current evidence strongly supports the role of FIS1 as a DRP1 receptor in mammalian cells. Key studies, including recent protein structural analyses (Egner et al., *J Biol Chem.* 2022 PMID: 36272645; Nolden et al., *J Biol Chem* 2023 PMID: 37866629) and cell biology investigations (Losón et al., *Mol Biol Cell* 2013 PMID: 23283981; Yoon et al. *Mol Cell Biol.* 2003 PMID: 12861026; Rios et al., *Nat Commun.* 2023 PMID: 37468472; Song et al., *Cell Mol Biol Lett* 2024 PMID: 38439028), collectively demonstrate the role of FIS1 as a DRP1 receptor in mammalian cells.

2. In Figure 1b, the use of transcriptomic datasets to correlate Drp1 expression levels with sarcoma lacks a rigorous experimental description. Patient ages, demographics and clinical conditions are crucial for interpreting such data. The criteria for defining high/low expression levels should also be clearly specified.

Response: We appreciate the reviewer's input. In the revised manuscript, we have included references to the clinical patient cohorts in the Experimental Procedures section.

Revised manuscript lines (367-368): Clinical characteristics of the patients are detailed in the respective publications (81,82).

The criteria for defining high/low expression levels were already described in the Experimental Procedures section, please see chapter 4.9, lines 368-370.

3. The experimental design for Figures 1c and 1d raises concerns; comparing different cell lines instead of pre- and post-therapy within the same cell line might lead to misleading conclusions due to variations in mitochondrial division machinery profiles among different cells.

Response: We agree with the reviewer that performing these experiments using a panel of isogenic, patient-matched pre-therapy and post-therapy cell lines would be ideal. However, we do not have access to such models. We therefore resorted to include models derived from tumors at diagnosis (pre-therapy) or at relapse after the therapy (post-therapy) to cover for the variability of tumor phenotypes observed in the clinics. The fact that this experimental design identified significant upregulation of phospho-DRP1 (S616) in two out of three post-therapy but none of the pre-therapy models implies this change is associated with clinically relevant therapy-resistant phenotypes. Indeed, results presented in Fig 2 and Fig 4 further validated that the levels of p-DRP1(S616) were responsive to chemotherapy treatment. While it might be interesting to test in the laboratory settings whether the long-term induction of chemotherapy resistance in pre-therapy cells would mimic the upregulation of p-DRP1(S616) observed in post-therapy models, we did not pursue these experiments as our following experiments with DRP1-knockdown cells clearly demonstrated that DRP1 does not provide viable therapeutic target in pediatric sarcoma cells. Thus, although the upregulation of p-DRP1(S616) may mark therapy-resistant cells, it does not mediate the resistance.

Nevertheless, in accordance with the reviewer's suggestions, we have amended the relevant part of the Results section and tuned down the interpretation of the analysis in Fig 1C,D – please see details in the response to the next comment below.

4. The claim that Drp1 S616 was regulated in 2 out of 3 post-therapy cell lines suggests the mechanism may not be universally applicable. Caution is advised in drawing conclusions, particularly in Figure 1 where the title asserts enhanced Drp1 activation post-therapy.

Response: We greatly appreciate the reviewer's feedback and agree that the data shown in Figure 1 needs to be interpreted more carefully. To ensure accurate interpretation, we have revised the relevant section of the results accordingly.

Revised manuscript lines (83-87): Notably, the activating phosphorylation of the mitochondrial fission mediator DRP1 at serine 616 (37) was strongly upregulated in 2 out of 3 post-therapy cell lines, indicating that enhanced DRP1 activation might be associated with a potential mechanism of adaptation to therapy-induced stress. However, to draw more robust conclusions, it will be necessary to expand the panel of tested cell lines in follow-up studies.

5. Figures 2c and 2d suffer from similar issues; the observed increase in Drp1 S616 levels under chemotherapy is inconsistent across different drug groups. The increasing is not detected in the VIN group. Moreover, total Drp1 levels increase in the TOP group, affecting the applicability of the conclusions drawn.

Response: We thank the reviewer for this feedback, but we respectfully disagree with the concerns raised about the inconsistency in Fig 2C and D. The levels of p-DRP1 (S616) were analyzed as the ratio of the phosphorylated form to the total protein form, following the standardized analysis of phosphorylated proteins (Ankenbauer et al., J Biol Chem. 2023, PMID: 37660914; Kalogeropoulou et al., *Biochem J.* 2022, PMID: 35950872, Blagoev et al. Nat Biotechnol, 2004). Upon vincristine (VIN) treatment, the p-DRP1(S616)/total DRP1 ratio was significantly increased compared to the untreated control, which is consistent with the effects of other analyzed drugs in Fig 2C and D. To avoid any confusion, we have provided a more detailed description of the method used for the analysis of phosphorylated protein forms in Experimental procedures section.

Revised manuscript lines (342-344): The levels of phosphorylated proteins (p-DRP1(S616), p-AMPK α (Thr172), p-ERK1/2(T202/YT204)) were analyzed as the ratio of the normalized phosphorylated form to the normalized total protein form (DRP1, AMPK α 1, and ERK1/2, respectively).

Additionally, we respectfully disagree with the reviewer's assertion that the total DRP1 levels increase in the TOP group. Upon topotecan (TOP) treatment, DRP1 levels were not increased – see Fig 2C and Source Data file, relevant blots shown below.

To avoid any confusion, we now present a different biological replicate of p-DRP1 (S616) and DRP1 for the DXR drug combination in the revised version of Fig 2C, as it better represents all biological replicates. Of course, to ensure transparency, all analyzed biological replicates are now included in the Source Data file for Fig 2.

6. Figure 3 lacks statistical analysis comparing the curves with and without rapamycin treatment, which is necessary for robust interpretation.

Response: We thank the reviewer for pointing this out to us. In response, we have added statistical analyses comparing the curves with and without rapamycin treatment to Figure S3. Similarly, we have included statistical analyses in Figure S4, which presents the chemoresistance analysis with and without bafilomycin A1 treatment.

7. In Figure 6, the authors conclude that depletion of Drp1 does not affect mitochondrial network morphology; however, the quality of the images is inadequate for reliable quantification, zoom field analysis is recommended. Additionally, the observed clustering of mitochondria in perinuclear regions in Drp1 KD cells is atypical and suggests potential issues with cell health.

Response: We appreciate the reviewer's suggestion to include zoom field analysis. We have added zoomed-in images in Fig 6 and Fig 7. To address the concerns raised about the quality of the mitochondrial imaging, we have newly included images of clone #3 in Fig S13 and Fig S15, alongside the images of clones #1 and #2 that were previously shown in the initially submitted version.

As demonstrated in Fig 5B,F, Fig S8, and Fig S9, the shDRP1 knockdown clones exhibited comparable growth rates and cell cycle profiles to the control clones, which, in our opinion, contradicts any potential issues with cell health. Moreover, after inspecting the raw data and images provided in Fig 6A, Fig 7A, Fig S13 and Fig S15, we do not share the reviewer's view that mitochondria change their distribution within DRP1-knockdown cells. Notably, such clustering would have affected results of mitochondrial morphology analysis. However, we did not detect any significant differences in mitochondrial network parameters among shCTRL and shDRP1 clones.

November 25, 2024

RE: Life Science Alliance Manuscript #LSA-2024-02870R

Dr. Jan Skoda
Masaryk University
Department of Experimental Biology
Kamenice 5
University Campus Bohunice, Bldg C13
Brno 63400
Czech Republic

Dear Dr. Skoda,

Thank you for submitting your revised manuscript entitled "Depleting chemoresponsive mitochondrial fission mediator DRP1 does not mitigate sarcoma resistance". We would be happy to publish your paper in Life Science Alliance pending final revisions necessary to meet our formatting guidelines.

- please be sure that the authorship listing and order is correct
- please upload your manuscript text as an editable doc file
- please add the Twitter handle of your host institute/organization as well as your own or/and one of the authors in our system
- please add a figure legend section to the main manuscript text that includes the legends for the main and supplementary figures and the table legends

A. FINAL FILES:

B. MANUSCRIPT ORGANIZATION AND FORMATTING:

Thank you for your attention to these final processing requirements. Please revise and format the manuscript and upload materials within 5 days.

Sincerely,

Reviewer #2 (Comments to the Authors (Required)):

The authors have addressed my concerns.

November 29, 2024

RE: Life Science Alliance Manuscript #LSA-2024-02870RR

Dr. Jan Skoda
Masaryk University
Department of Experimental Biology
Kamenice 5
University Campus Bohunice, Bldg C13
Brno 63400
Czech Republic

Dear Dr. Skoda,

Thank you for submitting your Research Article entitled "Depleting chemoresponsive mitochondrial fission mediator DRP1 does not mitigate sarcoma resistance". It is a pleasure to let you know that your manuscript is now accepted for publication in Life Science Alliance. Congratulations on this interesting work.

DISTRIBUTION OF MATERIALS:

Again, congratulations on a very nice paper. I hope you found the review process to be constructive and are pleased with how the manuscript was handled editorially. We look forward to future exciting submissions from your lab.

Sincerely,
